# *GenCP*: Towards Generative Modeling Paradigm of Coupled physics

**Tianrun Gao**[*][†][1,2] **Haoren Zheng**[*][†][1,3] **Wenhao Deng**[*][1] **Haodong Feng**[1]
**Tao Zhang**[†][1,4] **Ruiqi Feng**[1] **Qianyi Chen**[‡][1] **Tailin Wu**[‡][1]
[1]Department of Artificial Intelligence, School of Engineering, Westlake University;
[2]Department of Geotechnical Engineering, Tongji University;
[3]Global College, Shanghai Jiao Tong University;
[4]State Key Laboratory of Advanced Nuclear Energy Technology, Nuclear Power Institute of China;
`{gaotianrun, chenqianyi, wutailin}@westlake.edu.cn`

## Abstract

Real-world physical systems are inherently complex, often involving the coupling of multiple physics, making their simulation both highly valuable and challenging. Many mainstream approaches face challenges when dealing with decoupled data. Besides, they also suffer from low efficiency and fidelity in strongly coupled spatio-temporal physical systems. Here we propose *GenCP*, a novel and elegant generative paradigm for coupled multiphysics simulation. By formulating coupled-physics modeling as a probability modeling problem, our key innovation is to integrate probability density evolution in generative modeling with iterative multiphysics coupling, thereby enabling training on data from decoupled simulation and inferring coupled physics during sampling. We also utilize operator-splitting theory in the space of probability evolution to establish error controllability guarantees for this "conditional-to-joint" sampling scheme. We evaluate our paradigm on a synthetic setting and three challenging multiphysics scenarios to demonstrate both principled insight and superior application performance of *GenCP*. Code is available at this repo: github.com/AI4Science-WestlakeU/GenCP.

## 1 Introduction

Most real-world physical systems are governed by the intricate interplay of multiple physical processes spanning diverse disciplines (Yang et al., 2025). As a result, multiphysics problems are widely recognized as both fundamental and practically valuable. They arise across a broad range of applications, including aerospace engineering (Malikov et al., 2024; Hu et al., 2024), biological engineering (Pramanik et al., 2024; Gerdroodbary & Salavatidezfouli, 2025), and civil engineering (Wang et al., 2025; Wijesooriya et al., 2020). Despite their importance, accurate simulation of coupled physics remains notoriously difficult (McCabe et al., 2024), largely due to strong cross-physics interactions and the resulting high system complexity, which are far more challenging to model than in single-physics settings.

Numerical multiphysics simulation methods include tightly coupled methods, which achieve high fidelity by solving the entire system jointly (Knoll & Keyes, 2004; Yu et al., 2025), often incur prohibitive complexity and computational costs in real-world applications (Guo et al., 2025). Loosely coupled methods, which solve each physics field separately and iteratively exchange information until convergence, are more widely adopted, balancing practicality, efficiency, and acceptable precision (Löhner et al., 2006). However, both approaches suffer from requirements of extensive interdisciplinary expertise for solver development, and high computation demands (Fan & Wang, 2024).

Surrogate models (McCabe et al., 2024) and neural operators (Li et al., 2025), therefore, have emerged as an alternative to accelerate multiphysics simulation, given the limitations of numerical methods. Most of these models, however, rely heavily on coupled solutions as training data,

---

[*]Equal contribution. [†]Work done as an intern at Westlake University. [‡]Corresponding author.

making data acquisition at least five times more computationally expensive compared to using decoupled data (Degroote et al., 2008; Causin et al., 2005). To enable training on more accessible data from decoupled physics, surrogate models further borrow ideas from numerical solvers, such as the Gauss–Seidel framework (Milaszewicz, 1987) and perform ADMM-like (Deng et al., 2017) iterative inference to approximate the coupled solution (Lyu et al., 2025; Gao & Jaiman, 2024). Although improving efficiency and leveraging decoupled data, surrogate models struggle with complex spatiotemporal dynamics due to their limited ability to capture high-frequency, high-dimensional, and stochastic behaviors (Liang et al., 2024). The defect motivates emerging efforts in generative simulation (Fotiadis et al., 2025). Notably, existing generative approaches primarily focus on single-physics problems (Sun et al., 2023) or directly learn multiphysics solutions from coupled data (Park et al., 2021), largely overlooking the challenge of learning coupled physics from decoupled training data. A recent study (Zhang et al., 2025) explored embedding coupling iterations into every denoising step of diffusion models to enable coupled sampling. However, the study lacks a rigorous theoretical foundation or guarantees of reliability. Emerging generative approaches show promise for high-fidelity modeling but typically rely on coupled data or lack theoretical guarantees of reliability. The above challenges raise a critical **research question**: *How can we develop a framework that learns coupled physics from decoupled training data while ensuring high fidelity, efficiency, and reliability?*

To address the challenges, we introduce **Gen**erative **C**oupled **P**hysics Simulation (*GenCP*), a novel and principled paradigm for multiphysics simulation, which achieves decoupled training and coupled inference with "3H", combining high fidelity from generative modeling, high efficiency from our "coupling in flow" design, and high reliability guaranteed by numerical theory.

We begin by clarifying that the essence of solving coupled problems can be reformulated as modeling probability density evolution in functional spaces of physical fields. During training on decoupled data, we employ flow matching (Lipman et al., 2022) to learn the evolution of conditional probability, where each physical field evolves from noise to its target conditioned on the other one. During inference, we apply operator splitting in the functional space at each flow step, thereby merging conditional probability transitions into a coherent joint distribution transition. Our method physically corresponds to iteratively solving the coupled fields in the noisy latent space as they evolve toward the solution. Figure 1 illustrates the overall schematic of the *GenCP* paradigm.

Starting from the continuity equation of the joint distribution, we show that the flow matching (Lipman et al., 2022) sampling process can be reinterpreted with an operator-splitting scheme for probability density functions. We prove in a Hilbert space that our proposed joint inference method enjoys error controllability guarantees. On top of the theoretical foundation, we examine *GenCP* on both naive and complex settings. First, we design a synthetic dataset in 2D space to demonstrate the joint distribution sampling effect from learning the conditional distribution. Then, we evaluate *GenCP* on three challenging multiphysics settings and compare it against surrogate-based and other generative paradigms. Across these tasks, *GenCP* consistently outperforms existing approaches with error reducing ranging from 12.54% to 42.85%, and even exceeding 65% on certain metrics, demonstrating its effectiveness and robustness in coupled scenarios.

The contributions of our work include: (1) We reformalize generative coupled simulation by modeling it as probability density evolution in the functional space of the physical field, offering a principled view of the problem. (2) We propose *GenCP*, an efficient generative paradigm that achieve coupled inference with decoupled learning by introducing an operator splitting mechanism into the flow matching process. (3) We establish a theoretical guarantee by connecting the flow-matching continuity equation with operator splitting, and prove that our scheme enjoys controllable error bounds. (4) We conduct concrete evaluations on both synthetic setting and challenging multiphysical scenarios, where *GenCP* consistently outperforms surrogate and generative baselines.

## 2 RELATED WORK

In this section, we briefly review the related work on both numerical and learning-based coupling simulation, highlighting key insights as well as the gaps that motivate our approach. A full discussion is provided in Appendix A.

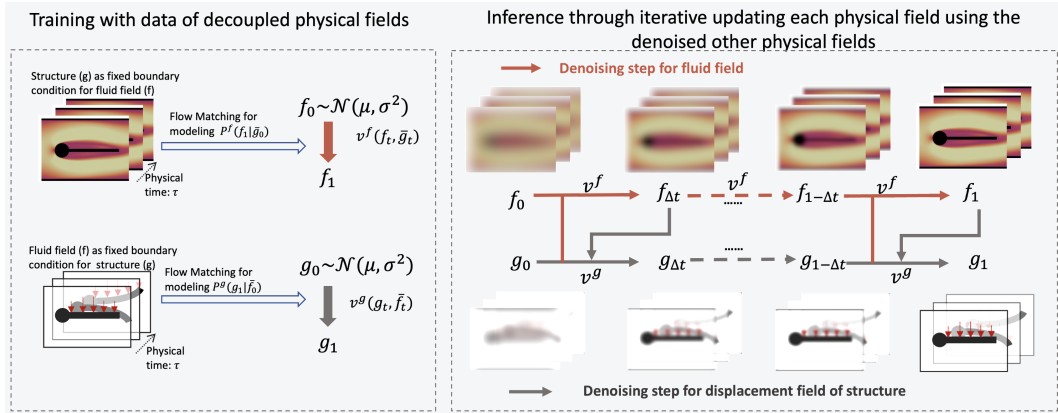

Figure 1: The schematic of *GenCP*. With a pretrained model with decoupled physics, *GenCP* could manage to achieve coupled physics during flow steps. Note that here we only use Lie-Trotter Splitting as a demonstration.

**Numerical simulation.** Classical numerical solvers are typically categorized into tightly coupled (monolithic) and loosely coupled (partitioned) schemes. Monolithic approaches enforce interface conditions exactly and ensure strong stability, but are computationally prohibitive (Ruan et al., 2021; Sun et al., 2024). Partitioned approaches are more efficient and easier to implement, though they are often prone to unstable convergence (MacNamara & Strang, 2016; Yue & Yuan, 2011; Ha et al., 2023; Fourey et al., 2017). To mitigate this trade-off, techniques such as operator splitting (Godunov & Bohachevsky, 1959; Strang, 1968; Trotter, 1959) offer theoretically grounded compromise between strictly monolithic and partitioned approaches.

**Learning-based simulation.** When coupled training data are available, neural operators have been extended to directly model multiphysics PDEs with enhanced cross-field communication, underscoring the importance of functional priors (Rahman et al., 2024; Li et al., 2025; Kerrigan et al., 2023). In settings where only decoupled data are accessible, surrogate models aim to reconstruct coupled solutions via iterative inference, yet often fail to capture high-frequency, high-dimensional dynamics (Sobes et al., 2021; Tiba et al., 2025). More recently, generative approaches have been explored to improve fidelity, but existing efforts either remain confined to single-physics scenarios or lack rigorous theoretical guarantees, limiting their applicability (Zhang et al., 2025; Sun et al., 2023).

## 3 METHOD

We organize the methodology into three parts. Section 3.1 formulates the coupled problem, introduces the weak continuity-equation perspective, and explains how decomposing the velocity field naturally leads to operator splitting as the basis for our method. Section 3.2 describes the time-parameterized linear interpolation used to produce instantaneous-velocity targets and the resulting training objectives. Section 3.3 presents coupled inference via operator splitting together with an informal error bound. Finally, Appendix B provides the precise assumptions and complete technical proofs of stability, consistency, and convergence.

### 3.1 PROBLEM REFORMULATION

Let $D \subset \mathbb{R}^d$ be a bounded open domain where the physical fields live. The first field is a function $f : D \to \mathbb{R}^{d_f}, f \in \mathcal{F}$, and the second field is $g : D \to \mathbb{R}^{d_g}, g \in \mathcal{G}$. For concreteness and implementation one may take $\mathcal{F} = L^2(D; \mathbb{R}^{d_f})$ and $\mathcal{G} = L^2(D; \mathbb{R}^{d_g})$, which are separable Hilbert spaces. The joint state is $u = (f, g)$ in the product space $\mathcal{U} := \mathcal{F} \times \mathcal{G}, \|u\|_{\mathcal{U}}^2 = \|f\|_{\mathcal{F}}^2 + \|g\|_{\mathcal{G}}^2$. We consider probability laws $\mu$ on $\mathcal{U}$ with finite second moment, denoted $\mu \in \mathcal{P}_2(\mathcal{U})$. The coupled evolution of the joint state is described by a family $\{\mu_t\}_{t \in [0,1]}$. In flow-matching, we seek a time-dependent velocity vector field $v(t, u) : [0, 1] \times \mathcal{U} \to \mathcal{U}$, such that $\{\mu_t\}$ is transported by $v$.

Fully coupled trajectories are typically unavailable. Instead we assume access to decoupled single-field solver data over a unit physical time step, where subscripts 0 and 1 denote the beginning and

end of that solver step in physical time (not to be confused with the normalized interpolation time $t \in [0, 1]$ used later for flow matching): $\mathcal{D}_f = \{(f_0, g_0) \mapsto (f_1, \bar{g})\}, \mathcal{D}_g = \{(f_0, g_0) \mapsto (\bar{f}, g_1)\}$. Here $\bar{g}$ means that the $g$-field is held fixed while $f$ evolves from $f_0$ to $f_1$, and $\bar{f}$ means that the $f$-field is frozen while $g$ evolves from $g_0$ to $g_1$. The two-field presentation is for clarity; *GenCP* can extend to $m$ fields by learning from decoupled datasets where each field evolves while others are frozen, and composing them in a cyclic order.

**Weak continuity equation.** We first recall the strong-form continuity equation. If the joint probability law $\mu_t$ admits a density $\rho_t(u)$ on the functional state space $\mathcal{U} = \mathcal{F} \times \mathcal{G}$, and if $\rho_t$ is sufficiently regular, then its evolution under a time-dependent velocity field $v(t, u)$ satisfies the strong-form PDE

$$\partial_t \rho_t(u) + \nabla_u \cdot \big(\rho_t(u)\, v(t, u)\big) = 0, \tag{1}$$

expressing conservation of probability mass as it is transported by $v$. In finite dimensions this PDE is well-defined, but for distributions over function spaces several obstacles arise: empirical measures coming from decoupled physics solvers are typically singular and do not admit a density $\rho_t$; the divergence operator $\nabla_u \cdot (\rho_t v)$ is not meaningful when $u$ represents functions rather than vectors; and even when densities exist, numerical or learned models cannot reliably approximate derivatives in infinite-dimensional spaces.

These limitations motivate the use of the *weak continuity equation* (Kerrigan et al., 2023), which defines the evolution of $\mu_t$ only through its action on smooth test functions $\varphi : \mathcal{U} \to \mathbb{R}$:

$$\int_0^1 \int_{\mathcal{U}} \Big(\partial_t \varphi(u) + \langle D\varphi(u), \, v(t, u)\rangle_{\mathcal{U}}\Big)\, d\mu_t(u)\, dt = 0, \tag{2}$$

where $v(t, u)$ is the time-dependent velocity vector field. Instead of differentiating the measure or its density, the weak form enforces that *all observables evolve consistently with the velocity field*. This provides a mathematically well-posed description of measure transport in infinite-dimensional settings such as $\mathcal{F} \times \mathcal{G}$.

**Decomposing the weak equation.** To exploit the decoupled datasets we decompose Because the weak formulation is linear in $v$, we obtain

$$\int_0^1 \int_{\mathcal{U}} \big(\partial_t \varphi(u) + (\mathcal{L}_f(t) + \mathcal{L}_g(t))\varphi(u)\big)\, d\mu_t(u)\, dt = 0,$$

where the component Liouville operators are

$$\mathcal{L}_f(t)\varphi(u) := \langle v^{(f)}(t, u), D\varphi(u)\rangle_{\mathcal{U}}, \qquad \mathcal{L}_g(t)\varphi(u) := \langle v^{(g)}(t, u), D\varphi(u)\rangle_{\mathcal{U}}.$$

Thus if we can learn $v_f$ and $v_g$ from $\mathcal{D}_f$ and $\mathcal{D}_g$, their sum recovers the generator of the joint weak evolution. The decomposition enables us to consider the weak continuity equations formed separately for each component velocity field. For the $f$-component: $\int_0^1 \int_{\mathcal{U}} \Big(\partial_t \varphi(u) + \langle D\varphi(u), v^{(f)}(t, u)\rangle_{\mathcal{U}}\Big) d\mu_t(u)\, dt = 0$. For the $g$-component: $\int_0^1 \int_{\mathcal{U}} \Big(\partial_t \varphi(u) + \langle D\varphi(u), v^{(g)}(t, u)\rangle_{\mathcal{U}}\Big) d\mu_t(u)\, dt = 0$.

**Lie–Trotter splitting.** The Lie–Trotter splitting scheme provides a principled mechanism for recombining the two learned partial dynamics during inference. Because $v = v^{(f)} + v^{(g)}$, the full coupled evolution can be approximated over a small time step $\tau$ by first evolving $f$ with $g$ held fixed (the flow induced by $v^{(f)}$), and then evolving $g$ with $f$ held fixed (the flow induced by $v^{(g)}$). In the limit of small $\tau$, this alternating update is consistent with the joint flow generated by $v^{(f)} + v^{(g)}$, which explains how separately learned conditional velocities can be merged into a coherent coupled inference procedure.

To formalize this, let $S_{t+h \leftarrow t}$ denote the observable propagator $(S_{t+h \leftarrow t}\varphi)(u) := \varphi(\Phi_{t+h \leftarrow t}(u))$, where $\Phi_{t+h \leftarrow t}$ is the flow map of the full velocity field $v$. For sufficiently small $h$, the propagator admits the first-order approximation

$$S_{t+h \leftarrow t} \approx S^{(g)}_{t+h \leftarrow t} \circ S^{(f)}_{t+h \leftarrow t},$$

where $S^{(f)}$ and $S^{(g)}$ correspond to the flows induced by $v^{(f)}$ and $v^{(g)}$, respectively. By duality, this translates to the measure-level operator splitting

$$\mu_{t+h} \approx \left( \Phi^{(g)}_{t+h \leftarrow t} \circ \Phi^{(f)}_{t+h \leftarrow t} \right)_{\#} \mu_t. \tag{3}$$

Iterating this update with step size $\tau$ yields the classical Lie–Trotter product formula used during inference (Trotter, 1959; Strang, 1968; Bátkai et al., 2011). This justifies recombining the separately learned conditional velocity fields through alternating partial flows, thereby recovering an approximation of the true coupled dynamics. Rigorous stability, consistency, and convergence results are deferred to Appendix B.

### 3.2 TIME-PARAMETERIZED LINEAR INTERPOLATION AND TRAINING

We now give a fully constructive procedure that produces instantaneous-velocity targets from decoupled data.

**Reference distributions.** Choose tractable reference measures $\pi_{\mathcal{F}}$ on $\mathcal{F}$ and $\pi_{\mathcal{G}}$ on $\mathcal{G}$. In practice, these may be simple Gaussian priors defined on the finite discretization of $D$.

**Learning the velocity for the field.** Sample $(f_1, \bar{g}) \sim \mathcal{D}_f$ and independent references $z_f \sim \pi_{\mathcal{F}}$, $z_g \sim \pi_{\mathcal{G}}$. For $t \in [0,1]$ define the linear interpolants $f_t = (1-t)z_f + tf_1$ and $g_t = (1-t)z_g + t\bar{g}$. The instantaneous derivative along this interpolation is vector

$$\frac{df_t}{dt} = v_f = f_1 - z_f.$$

We use $(f_t, g_t, t)$ as input to the conditional velocity model for $f$ and take $v_f$ as target. This setup yields a random supervisory signal whose expectation conforms to the conditional velocity stipulated in the weak continuity equation. Symmetrically, sample $(\bar{f}, g_1) \sim \mathcal{D}_g$ with independent references $z'_f, z'_g$ and set $f_t = (1-t)z'_f + t\bar{f}$, $g_t = (1-t)z'_g + tg_1$. The vector is given by $\frac{dg_t}{dt} = v_g = g_1 - z'_g$. Then $(f_t, g_t, t)$ is the input and $v_g$ is the target for the conditional velocity of $g$.

**Learning objectives.** Parameterize two operator-valued models $\hat{v}_f(f, g, t; \theta_f)$ and $\hat{v}_g(f, g, t; \theta_g)$ that map inputs $(f_t, g_t, t)$ to elements of $\mathcal{F}$ and $\mathcal{G}$, respectively. Minimize the mean-square losses

$$\mathcal{L}_f(\theta_f) = \mathbb{E}_{t, (f_1, \bar{g}) \sim \mathcal{D}_f, z_f, z_g} \left[ \| v_f - \hat{v}_f(f_t, g_t, t; \theta_f) \|^2_{\mathcal{F}} \right], \tag{4}$$

$$\mathcal{L}_g(\theta_g) = \mathbb{E}_{t, (\bar{f}, g_1) \sim \mathcal{D}_g, z'_f, z'_g} \left[ \| v_g - \hat{v}_g(f_t, g_t, t; \theta_g) \|^2_{\mathcal{G}} \right]. \tag{5}$$

The two training procedures use different decoupled datasets, which match typical solver outputs. The specific training algorithm can be found in Appendix F.1.

### 3.3 COUPLED INFERENCE VIA OPERATOR SPLITTING

**Coupled inference.** After training we obtain learned conditional velocity estimators $\hat{v}_f$ and $\hat{v}_g$. We then recombine their induced flows using Lie–Trotter splitting to produce coupled samples. Concretely, let $\Phi^{(A)}_s$ be the flow map obtained by integrating $\hat{v}^{(A)} = (\hat{v}_f, 0)$ and $\Phi^{(B)}_s$ the flow map for $\hat{v}^{(B)} = (0, \hat{v}_g)$. For a step size $\tau = 1/N$, the Lie–Trotter push-forward update is given by Eq. 3, which at the state level corresponds to the explicit alternating update:

---

**Algorithm 1** Coupled Sampling via Lie–Trotter Splitting

---

1: Input: initial $u_0 = (f_0, g_0)$, trained operators $\hat{v}_f, \hat{v}_g$, steps $N$
2: $u \leftarrow u_0$
3: **for** $k = 0$ to $N - 1$ **do**
4:     $t \leftarrow k/N$
5:     $f \leftarrow f + \tau \hat{v}_f(f, g, t; \theta_f)$
6:     $g \leftarrow g + \tau \hat{v}_g(f, g, t; \theta_g)$
7:     $u \leftarrow (f, g)$
8: **end for**
9: **return** $u$

---

**Theoretical guarantee of an acceptable error bound.** The two error sources of the learned splitting scheme are the splitting discretization and the learning approximation. Under the stability and regularity hypotheses stated precisely in Appendix B, one can show the following bound.

**Theorem 3.1** (Informal error bound). *Let $\mu_1$ denote the true terminal measure at $t = 1$ generated by the unknown joint velocity $v$. Let $\mu_1^{(\tau,\mathrm{learn})}$ denote the terminal measure obtained by applying the learned Lie–Trotter composition with step size $\tau$ using $\hat{v}_f, \hat{v}_g$. If the learned fields uniformly approximate the true conditional velocities on the region visited by the dynamics with errors at most $\varepsilon_f, \varepsilon_g$, then*

$$W_1\big(\mu_1^{(\tau,\mathrm{learn})}, \mu_1\big) \leq C_{\mathrm{stab}}\big(\tau + \varepsilon_f + \varepsilon_g\big),$$

*where $W_1$ is the Wasserstein-1 distance and $C_{\mathrm{stab}}$ depends on Lipschitz and growth constants for the flows.*

The term $\tau$ comes from the first-order nature of Lie–Trotter splitting, meaning the error introduced by time-splitting scales linearly with the step size. The terms $\varepsilon_f, \varepsilon_g$ capture how well the learned conditional velocities approximate the true conditionals. Thus, improving either time resolution or regression accuracy reduces the total error.

Precise assumptions, the stability lemma for propagator composition, the infinitesimal consistency of the split generator, and the detailed convergence proof are collected in Appendix B. Those technical statements track which Lipschitz and growth conditions are needed and make the informal bound above rigorous.

## 4 EXPERIMENTS

In this section, we aim to answer the following 3 questions: **(1)** In the purely mathematical setting of probability distributions, can *GenCP* enable training two separate flow matching models on datasets describing conditional distributions, and then directly combine them at inference to sample from the joint distribution? **(2)** Beyond the mathematical case, can *GenCP* maintain strong performance (effectively modeling coupling behaviors at inference with only decoupled training) in complex and high-dimensional coupled simulation problems? **(3)** In addition to simulation accuracy, how does the inference efficiency of *GenCP* compare with existing paradigms? Next, question (1) is addressed in Section 4.2, and questions (2) and (3) are answered in Section 4.3 and Section 4.4.

### 4.1 EXPERIMENTAL SETUP

We evaluate *GenCP* on four representative cases: a simple 2D distribution, designed to directly illustrate our conditional-to-joint sampling paradigm, two canonical FSI benchmarks (Double-Cylinder and Ture-Hron) and a complex nuclear-thermal setting. To assess the capability of *GenCP*, we compare it against two alternative paradigms for modeling coupled physics, each with two neural operator backbones (CNO and FNO*[0]). The two alternative paradigms represent (i) a surrogate model–based Picard-iteration approach and (ii) the M2PDE paradigm.

Across all tasks, training is conducted exclusively on decoupled datasets, following Algorithm 2. In practice, this means we model the distributional evolution of one variable while treating the other as a known control condition. A subset of the decoupled data is reserved as a validation set to evaluate learning on the decoupled setting, and the coupled data serve as the test set to assess the coupling performance. Further details on the datasets used can be found in Appendix D.

At inference time, *GenCP* performs coupled simulation by alternately evolving from noise to the target distribution using the learned conditional distributions via operator splitting. Specifically, we adopt the Lie–Trotter splitting method, as shown in Algorithm 1, to achieve a balanced trade-off between precision and computational cost.

For comparison among the paradigms, the surrogate-based paradigm trains a neural network that end-to-end maps system inputs to outputs on decoupled data and then applies Picard iteration at inference to approximate the coupled solution. The M2PDE paradigm instead trains a diffusion

---

[0]FNO* denotes the FNO framework with SiT as its internal model, making the backbone more expressive for our learning objectives while retaining its functional properties.

Table 1: Comparison of statistical distances between the target joint distributions and those generated from different paradigms.

|  | Easy Distribution | | Complex Distribution | |
| --- | --- | --- | --- | --- |
|  | *GenCP* | M2PDE | *GenCP* | M2PDE |
| $W_1$ | **0.4366** | 0.5177 | **0.4928** | 25450.5442 |
| MMD | **0.0095** | 0.0141 | **0.0053** | inf |
| Energy Distance | **0.0411** | 0.0625 | **0.0061** | 332.3619 |

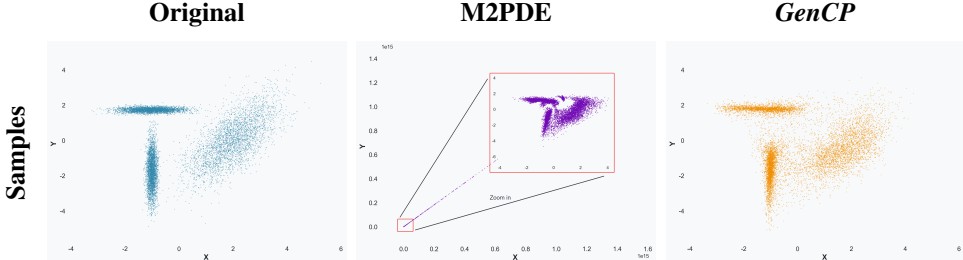

Figure 2: Visualization of sampling results on complex distribution with different paradigms in the synthetic setting. In order to examine behavior except outlier sampling, the visualized results of M2PDE also include a zoomed-in view on the same coordinate region as the other views.

model on each decoupled dataset, using the DDIM estimation on the denoised result of one field as the condition to denoise the other. Detailed implementation settings and pseudo-code for baseline paradigms are provided in Appendix F.

Full implementation details (including backbone architectures, sampling paradigms, and training configurations) are provided in Appendix. As an additional contribution, we also open-source the dataset used in this study to support further method development in coupled simulation.

## 4.2 SYNTHETIC SETTING ON 2D DISTRIBUTION

To intuitively demonstrate the core idea of *GenCP*, that learning $p(x \mid y)$ and $p(y \mid x)$ on decoupled samples suffices for direct joint sampling of $p(x, y)$ during inference, we first investigate it on the synthetic dataset. We note that M2PDE (Zhang et al., 2025) also tried to sample on the joint distribution with a generative model based on training with conditional data. So here we compare our performance with M2PDE.

We design two synthetic distributions as synthetic dataset: an "Easy distribution" based on simple Gaussian mixture, and a "Complex distribution" with multiple Gaussian components arranged in a multimodal pattern. To obtain conditional information while avoiding direct joint sampling, we generate two complementary datasets for training that separately focus on $p(y|x)$ and $p(x|y)$ with perturbed marginal for $x$ and $y$, respectively. This setup ensures that models are trained only on conditionals, while the underlying target joint structure remains nontrivial. Further details of the synthetic setting could be seen in Appendix D.

We visualize the sampling effect in Figure 2 and report quantitative metrics, including Wasserstein-1 distance, MMD, and Energy distance between the target distribution and the generated one in Table 1. The distributional distance results show that our paradigm slightly outperforms the baseline on simple distribution and achieves a substantial advantage on more complex distribution, maintaining stable performance while the baseline degrades severely. These results demonstrate that *GenCP* can successfully recover the true joint distribution with only conditional training, closely matching the ground truth both visually and in distributional metrics. In contrast, the baseline method performs poorly: it not only fails to capture the main body of the distribution, but also exhibits strong instability that leads to frequent sampling of outliers. This weakness arises from its iterative-to-convergence design, as well as the error accumulation induced from using intermediate estimates as conditions, ultimately causing mode collapse or drift, especially in complex distributions.

### 4.3 APPLICATION TO FSI SCENARIOS

We next evaluate GenCP on two fluid-structure coupling tasks, which exemplify real multiphysics coupling with strong bidirectional feedback. Both settings are widely used benchmarks for FSI and present significant challenges: strongly coupled fields, high-dimensional and high-frequency features, as well as nonlinear feedback loops. The FSI tasks here are formulated as follows: given 3 known steps of both fields, the goal is to predict the couple dynamics over the next 12 steps, using a model trained only with decoupled data.

In both settings, we describe fluid behavior using velocity fields ($u$, $v$) and a pressure field ($p$), while structural dynamics are represented by a signed distance field (SDF). Thus, the multiphysics representation of the FSI problem can be composed into a four-channel field defined on the same computational domain. Through this careful design, we avoid the common difficulty encountered when representing fluid-structure interface in normal FSI algorithms, such as Arbitrary Lagrangian-Eulerian (ALE) (Korobenko et al., 2018) and Immersed Boundary Method (IBM) (Tian, 2014).

**Turek-Hron.** Turek-Hron is a classical FSI case: a rigid cylinder is placed in cross-flow, followed by a flexible beam that bends under the influence of vortices shed in the wake. This setup is also widely used as a standard for evaluating numerical FSI solvers, making it a stringent test of fidelity. We conduct training and inference following the procedure described in the previous section.

Table 2 compares the relative L2 norm errors and inference costs of the three inference paradigms across two backbone models. On both FNO* and CNO backbones, *GenCP* consistently achieves better test accuracy on coupled data than both surrogate-based and M2PDE paradigms. With the FNO* backbone, the average error across four fields is about 26.77% lower than the best-performing baseline, and with the CNO backbone, the error is about 12.54% lower. Furthermore, the CNO backbone even achieves results on SDF that are close to those of Joint Training. In terms of efficiency, our paradigm demonstrates an extremely pronounced advantage: thanks to the operator-splitting design that embeds coupling directly into the flow matching process, *GenCP* requires only 10 sampling steps to generate accurate coupled solutions. This design allows *GenCP* to simultaneously maintain accuracy while significantly improving inference efficiency compared to surrogate–based explicit iterations or diffusion-based iterative coupling condition on estimated fields.

Table 2: The relative L2 norm error and inference time of different methods on the Turek-Hron setting.

| Rel L2 Norm | Validation on Decoupled Data | | | | Test on Coupled Data | | | | Inference Time |
|---|---|---|---|---|---|---|---|---|---|
| Field | $u$ | $v$ | $p$ | SDF | $u$ | $v$ | $p$ | SDF | |
| Joint Training | / | / | / | / | 0.0088 | 0.03441 | 0.0544 | 0.0079 | / |
| M2PDE-FNO* | 0.0452 | 0.1582 | 0.1444 | 0.0206 | 0.0590 | 0.2415 | 0.2474 | 0.2482 | 277.20s |
| Surrogate-FNO* | 0.0181 | 0.0756 | 0.0953 | 0.0087 | 0.0550 | 0.2257 | 0.2553 | 0.0112 | 93.20s |
| Our *GenCP*-FNO* | 0.0091 | 0.0471 | 0.0548 | 0.0069 | **0.0396** | **0.1678** | **0.1897** | **0.0081** | **19.50s** |
| M2PDE-CNO | 0.0497 | 0.2075 | 0.3466 | 0.03937 | 0.05769 | 0.2809 | 0.4087 | 0.0390 | 347.00s |
| Surrogate-CNO | 0.0204 | 0.0994 | 0.1526 | 0.0205 | 0.0469 | 0.1888 | 0.2278 | 0.0242 | 300.25s |
| Our *GenCP*-CNO | 0.0150 | 0.0626 | 0.1187 | 0.0152 | **0.0388** | **0.1821** | **0.2166** | **0.0183** | **16.25s** |

Figure 3 provides a visual comparison of the coupled solutions for each field across different paradigms using FNO* as the backbone. As shown, our method closely matches the ground truth, with error maps confirming its ability to preserve both global patterns and the high-frequency details induced by fluid-structure interactions, and it clearly outperforms the baseline methods. In particular, examining the reconstructed SDF field highlights that, although the surrogate-based method reports relatively low error values in Table 2, it fundamentally fails to model the oscillatory bending dynamics of the beam. By contrast, our *GenCP*, while trained only on decoupled data, is the only method that can successfully capture these bending effects, which are intrinsic to true coupling. In fact, the seemingly low error of surrogate models arises from their deterministic end-to-end prediction. In contrast, our probabilistic modeling approach accounts for both mode errors and stochastic noise, explaining why our visual results appear far superior to the baselines, even though the quantitative metrics show modest improvements.

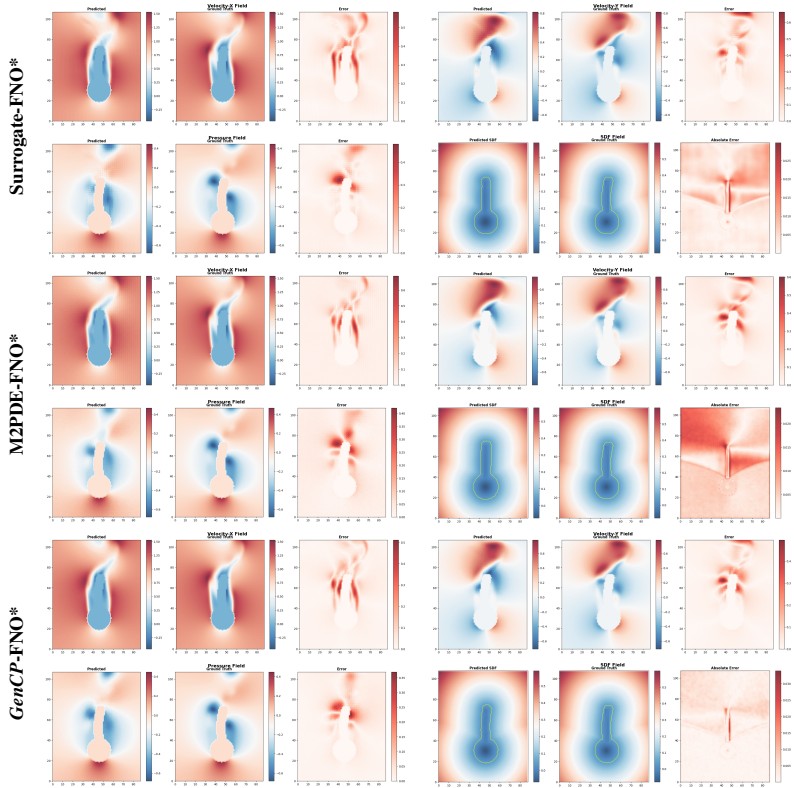

Figure 3: The visualization of results from different paradigms using FNO* as backbone on Turek-Hron setting. (Note that the structural boundary shown in the fluid field visualization is masked from the ground truth and does not represent actual structural deformation.)

Table 3: The relative L2 norm error of different methods on the Double Cylinder setting.

| Rel L2 Norm | Validation on Decoupled Data | | | | Test on Coupled Data | | | |
|---|---|---|---|---|---|---|---|---|
| Field | $u$ | $v$ | $p$ | SDF | $u$ | $v$ | $p$ | SDF |
| Joint Training | / | / | / | / | 0.0110 | 0.0387 | 0.0786 | 0.0034 |
| M2PDE-FNO* | 0.0128 | 0.0414 | 0.2238 | 0.0065 | 0.0571 | 0.2583 | 0.8879 | 0.0152 |
| Surrogate-FNO* | 0.0189 | 0.0625 | 0.0681 | 0.0041 | 0.0717 | 0.3058 | 0.4979 | 0.0196 |
| Our *GenCP*-FNO* | 0.0098 | 0.0441 | 0.0525 | 0.0060 | **0.0522** | **0.2397** | **0.3987** | **0.0061** |
| M2PDE-CNO | 0.0295 | 0.2072 | 0.6468 | 0.0156 | 0.0379 | 0.2694 | 0.9966 | 0.0179 |
| Surrogate-CNO | 0.0364 | 0.1078 | 0.2060 | 0.0067 | 0.0612 | 0.2058 | 1.0722 | 0.0080 |
| Our *GenCP*-CNO | 0.0141 | 0.0474 | 0.1753 | 0.0058 | **0.0279** | **0.1150** | **0.6208** | **0.0055** |

**Double cylinder.** To evaluate the performance of our method on more strongly coupled problems, we construct a more challenging FSI scenario by increasing the time step. Specifically, we adopt a double-cylinder configuration, in which one cylinder is fixed in a cross-flow while the other is attached to a linear spring and allowed to oscillate vertically with respect to the inlet. This setup induces strong bidirectional feedback between vortex shedding and structural oscillation. The training and inference procedures are identical to those described previously and are not repeated here.

The error analysis of *GenCP* compared with baseline paradigms is quantified in Table 3. The trends are consistent with those observed in the Turek-Hron case. Evidently, our method significantly outperforms baseline methods across all physical fields, with average error reductions of approximately 34.44% on FNO* backbone and up to 42.85% on CNO backbone. These results demonstrate that *GenCP* exhibits even greater advantages over baselines in strongly coupled problems.

In addition, we also report results obtained by training flow matching models directly on coupled data, *i.e.*, learning the joint distribution. Theoretically, this approach represents the upper bound

of our method. In other words, the discrepancy between the results of joint training and those of training on decoupled data reflects the error introduced by "conditional learning to joint sampling". Although the error values from joint training remain lower, the performance of *GenCP* is already very close to this upper bound compared to other baselines, highlighting the value of our proposed paradigm. For additional visualization results and analyses, please refer to Appendix E.

## 4.4 APPLICATION TO NT COUPLING SCENARIOS

To further demonstrate the scalability and generality of *GenCP*, we include an additional experiment on a Nuclear–Thermal Coupling problem. Here *GenCP* can be seamlessly applied to this more complex multiphysics scenario, involving scaling to additional physical fields, and stronger or more intricate coupling mechanisms.

The NT coupling system requires jointly solving neutron diffusion, solid heat conduction, and fluid heat transfer equations, while accounting for (i) the negative thermal feedback between neutron physics and temperature, (ii) the unidirectional influence of the fluid field on the neutron field through temperature, and (iii) the strong interface coupling between the fuel and the coolant. The geometry and coupling configurations of this system are illustrated in Figure 4. In this setting, our goal is to predict the system's transient evolution of coupling physics under different boundary conditions (neutron flux and fuel temperature measurement at the left side), with learning only on completely decoupled data.

The visualizations of the experimental results with three paradigms are provided in Figure 8. As shown in the Table 4, *GenCP* consistently outperforms both baselines across the neutron, fluid, and solid fields. Specifically, using FNO* as the backbone, *GenCP* achieves an average error reduction of 49.9% relative to M2PDE and 51.7% relative to the Surrogate paradigm. With the CNO backbone, the improvements are even more substantial, with error reductions of 58.2% over M2PDE and 78.8% over Surrogates. These results demonstrate that our method significantly surpasses existing baselines on "decoupled-training to coupled-inference" tasks in complex multiphysics settings.

To further investigate the source of errors, we also report the validation error on the coupled dataset, obtained by models evaluated using the ground-truth auxiliary coupled fields as known conditions. This error corresponds to the approximation error $\epsilon$ in Theorem 3.1, while the discrepancy between this prediction and the actual coupled inference result reflects the splitting error $\tau$. The results indicate that, consistent with our theoretical guarantees, the error introduced by our conditional-to-joint sampling method is minimal.

Table 4: The relative L2 norm error of different methods on the Nuclear-Thermal Coupling setting.

| Rel L2 Norm | Decoupled validation on decoupled data | | | Decoupled validation on coupled data | | | Coupled test on coupled data | | |
|---|---|---|---|---|---|---|---|---|---|
| Field | Neutron | Fuel | Fluid | Neurton | Fuel | Fluid | Neurton | Fuel | Fluid |
| Our *GenCP*-FNO* | 0.0022 | 0.0006 | 0.0038 | 0.0081 | 0.0371 | 0.0364 | **0.0085** | **0.0364** | **0.0270** |
| Surrogate-FNO* | 0.0086 | 0.0014 | 0.0032 | 0.0140 | 0.0167 | 0.0767 | 0.0149 | 0.0576 | 0.1095 |
| M2PDE-FNO* | 0.0052 | 0.0014 | 0.0018 | 0.0082 | 0.0085 | 0.0237 | 0.0136 | 0.1237 | 0.0463 |
| Our *GenCP*-CNO | 0.0024 | 0.0005 | 0.0110 | 0.0047 | 0.0083 | 0.0303 | **0.0044** | **0.0105** | **0.0330** |
| Surrogate-CNO | 0.0046 | 0.0007 | 0.0082 | 0.0073 | 0.0044 | 0.0567 | 0.0130 | 0.0553 | 0.3086 |
| M2PDE-CNO | 0.0053 | 0.0016 | 0.0092 | 0.0084 | 0.0017 | 0.0236 | 0.0164 | 0.0646 | 0.0401 |

## 5 CONCLUSION

In this work, we tackle the fundamental challenges of coupled simulation, where efficiency, fidelity, and the ability to leverage decoupled data remain major bottlenecks. To address these challenges, we reformulated the problem from a functional probabilistic evolution perspective, recognizing that the essence of coupling is sampling from the joint distribution via learned conditional distributions. Building on this insight, we have introduced *GenCP*, a principled generative paradigm that enables decoupled training and coupled inference in flow matching, achieving substantially higher accuracy and efficiency than existing methods across three distinct scenarios. Moreover, by applying operator-splitting theory to the probability density evolution process, we established a provable error bound for this conditional-to-joint sampling scheme, providing *GenCP* with a solid theoretical foundation.

Looking ahead, we believe this work paves the way toward probabilistically principled and practically scalable generative paradigms for coupled simulation, uniting theoretical guarantees with real-world applicability.

ETHICS STATEMENT

This work proposes a generative modeling paradigm for coupled simulation using synthetic and simulated data and provides theoretical guarantee for it. All datasets are collected without involving sensitive information.

REPRODUCIBILITY STATEMENT

The code is available at github.com/AI4Science-WestlakeU/GenCP. We provide a unified and modular code framework, together with scripts for reproducing all experiments. We will also release the data and checkpoints with full documentation to ensure transparency and reproducibility.

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

## A  RELATED WORK

### A.1  NUMERICAL COUPLING SIMULATION

Traditional numerical simulation for coupling physics mainly target at solving major difficulties about ensuring accuracy, stability, and efficiency (Felippa et al., 2001; Hou et al., 2012). Classical numerical solvers distinguish between tightly coupled (monolithic) and loosely coupled (partitioned) schemes. Tightly coupled solvers assemble fluid and solid equations into a coupled system, enforcing interface conditions exactly and offering strong stability, but at the expense of high computational cost. Ruan et al. (2021) introduce a monolithic framework that accurately handles momentum transfer at liquid–solid–air interfaces. Sun et al. (2024) proposed an immersed multi-material arbitrary Lagrangian–Eulerian method to realize a monolithic coupling framework.

Loosely coupled solvers, in contrast, partition system into subproblems for more practical solution, while they may suffer from unstable convergence. Besides, equipped with numerical techniques such as operator splitting (MacNamara & Strang, 2016) and Picard iteration (Yue & Yuan, 2011), they typically enjoy theoretical guarantees on acceptable error bounds. Representative developments include improvements in numerical integration strategies, such as semi-implicit coupling schemes (Ha et al., 2023) and time-adaptive partitioned methods (Bukač et al., 2023), as well as advances in representation techniques, such as coupling SPH with FEM (Fourey et al., 2017), and a partitioned explicit Lagrangian FEM tailored for highly nonlinear multiphysics problems (Meduri et al., 2018).

More specifically about operator splitting, it provides a theoretically grounded compromise between strictly monolithic and partitioned approaches. By decomposing the global operator into sub-operators advanced sequentially, splitting schemes exploit sub-solvers while maintaining control over stability and error (Godunov & Bohachevsky, 1959; Strang, 1968; Trotter, 1959). Extensive surveys have confirmed that Lie-Trotter and Strang compositions remain central tools in multiphysics applications (Blanes et al., 2024; Bukač et al., 2014; Čanić et al., 2020).

### A.2  LEARNING-BASED SIMULATION

Existing learning-based approaches for multiphysics simulation have also basically evolved along two categories: tightly coupled (monolithic) and loosely coupled (partitioned) methods. The first

category leverages neural operators to directly simulate multiphysics PDEs. CANO (Rahman et al., 2024) introduced codomain attention to enhance information exchange across coupled PDE outputs, demonstrating its potential for improving generalizability in multiphysics problems. COMPOL (Li et al., 2025) further incorporated explicit cross-field interaction mechanisms and gained notably increased accuracy. Kobayashi et al. (2025) systematically examined the relationship between operator architectures and the coupling strength of multiphysics. Collectively, these works underscore the importance of embedding functional priors (Kerrigan et al., 2023) for multiphysics simulation.

The second category employs AI models to learn decoupled physics in systems and then recomposes them to obtain coupled solutions (Sobes et al., 2021). In nuclear engineering, AI methods have been proposed for modeling of complex thermo–hydraulic–neutronic interactions (Huang et al., 2025). For unsteady problems, Tiba et al. (2025) incorporated reduced-order models for enabling better convergence and efficiency. These studies demonstrate the effectiveness of surrogate models in accelerating coupled simulations while keeping accuracy. More recently, generative models have also been explored for coupled simulation (Zhang et al., 2025), like applying compositional generation to capture joint dynamics across multiple conditional models. However, its lack of theoretical guarantees limits its applicability in strongly coupled scenarios.

## B  DEFINITIONS AND PROOFS

### B.1  OPERATOR SPLITTING SCHEME: ASSUMPTIONS AND CONVERGENCE GUARANTEES

This appendix provides the operator-splitting analysis underlying the error control bound in Section 3.3.

**Assumptions.**  To ensure the well-posedness of the dynamics and convergence of the splitting scheme, we impose the following:

(A1) *Local Lipschitz and growth.* For every $R > 0$, there exists $L_R > 0$ such that for all $t \in [0, 1]$ and $u, \tilde{u} \in \mathcal{U}$ with $\|u\|_{\mathcal{U}}, \|\tilde{u}\|_{\mathcal{U}} \leq R$,

$$\|v(t, u) - v(t, \tilde{u})\|_{\mathcal{U}} \leq L_R \|u - \tilde{u}\|_{\mathcal{U}}, \qquad \|v(t, u)\|_{\mathcal{U}} \leq C_1 \|u\|_{\mathcal{U}} + C_2.$$

This ensures local existence and uniqueness of solutions.

(A2) *Well-posedness of subproblems.* For every initial state $u \in \mathcal{U}$, the A- and B-subproblems (evolving $f$ with $g$ frozen, and $g$ with $f$ frozen) are well-posed, generating evolution families $U_A(t, s)$ and $U_B(t, s)$. They satisfy:

  (i) *Existence*: for any starting time $s$, there exists a dense subspace $Y_s$ on which classical solutions exist.

  (ii) *Uniqueness*: each $y \in Y_s$ generates a unique solution.

  (iii) *Continuous dependence*: solutions depend continuously on $(s, y)$, uniformly on compact intervals.

  (iv) *Exponential boundedness*: there exist constants $M \geq 1$, $\omega \in \mathbb{R}$ such that

$$\|u_s(t, y)\| \leq M e^{\omega(t-s)} \|y\|, \quad \forall t \geq s.$$

(A3) *Accuracy of learned vector fields.* On the bounded state set $\mathcal{B}$ visited during evolution, the learned vector fields $\hat{v}_f, \hat{v}_g$ satisfy uniform approximation bounds

$$\sup_{u \in \mathcal{B}} \|v_f(t, u) - \hat{v}_f(t, u)\|_{\mathcal{F}} \leq \varepsilon_f, \quad \sup_{u \in \mathcal{B}} \|v_g(t, u) - \hat{v}_g(t, u)\|_{\mathcal{G}} \leq \varepsilon_g,$$

and are Lipschitz in $u$ with a uniform constant $L_{\mathrm{model}}$.

**Lie–Trotter splitting.**  With $v = v^{(A)} + v^{(B)}$, define the Liouville operators from the weak continuity equation

$$\mathcal{L}_A(t)\varphi(u) = \langle v_f(t, f, g), D_f \varphi(u) \rangle_{\mathcal{F}}, \qquad \mathcal{L}_B(t)\varphi(u) = \langle v_g(t, f, g), D_g \varphi(u) \rangle_{\mathcal{G}},$$

so that $\mathcal{L}(t) = \mathcal{L}_A(t) + \mathcal{L}_B(t)$. The splitting scheme evolves measures as

$$\mu_{t_{k+1}}^{(\tau)} = \left( \hat{\Phi}_{t_{k+1} \leftarrow t_k}^{(B)} \circ \hat{\Phi}_{t_{k+1} \leftarrow t_k}^{(A)} \right)_{\#} \mu_{t_k}^{(\tau)}, \qquad \tau = 1/n,$$

where $\hat{\Phi}^{(A)}$ and $\hat{\Phi}^{(B)}$ denote flows generated by the learned fields $\hat{v}_f, \hat{v}_g$. The global approximation over $[0, 1]$ is the $n$-fold composition

$$\mu_1^{(\tau)} = \big(\hat{\Phi}_\tau^{(B)} \circ \hat{\Phi}_\tau^{(A)}\big)_{\#}^n \mu_0.$$

**Stability.** Assume there exist constants $M \geq 1$, $\omega \in \mathbb{R}$ such that for any partition $\{t_k\}$ and corresponding propagators $\mathcal{S}^{(A)}, \mathcal{S}^{(B)}$,

$$\Big\| \prod_{k=0}^{n-1} \mathcal{S}_{t_{k+1}\leftarrow t_k}^{(B)} \circ \mathcal{S}_{t_{k+1}\leftarrow t_k}^{(A)} \Big\|_{\mathcal{L}(B)} \leq Me^\omega,$$

for a Banach space $B$ of observables controlling sup-norms and gradients.

**Consistency.** The scheme is consistent:

$$\lim_{h \to 0} \frac{\mathcal{S}_{t+h\leftarrow t}^{(B)} \circ \mathcal{S}_{t+h\leftarrow t}^{(A)}\varphi - \varphi}{h} = \mathcal{L}(t)\varphi,$$

for any $\varphi \in \mathcal{T}$. This follows from the Taylor expansion of the A- and B-flows on finite-dimensional projections.

**Convergence.** The local splitting error is

$$\mathcal{R}_h(\varphi; t, u) := \frac{\mathcal{S}_{t+h\leftarrow t}^{(B)} \circ \mathcal{S}_{t+h\leftarrow t}^{(A)}\varphi(u) - \varphi(u)}{h} - \mathcal{L}(t)\varphi(u),$$

which is $O(h)$ under regularity conditions. Thus, the global error is first order $O(\tau)$ when using exact vector fields. With learned fields, an additional error $\varepsilon_f + \varepsilon_g$ appears. Combining with stability yields the Wasserstein bound

$$W_1(\mu_1^{(\tau,\text{learn})}, \mu_1) \leq C_{\text{stab}}\big(\tau + \varepsilon_f + \varepsilon_g\big).$$

## B.2 PROOF OF STABILITY

The goal is to show that the composition of the propagators $\mathcal{S}_{t_{k+1}\leftarrow t_k}^{(A)}$ and $\mathcal{S}_{t_{k+1}\leftarrow t_k}^{(B)}$ satisfies the bound:

$$\Big\| \prod_{k=0}^{n-1} \mathcal{S}_{t_{k+1}\leftarrow t_k}^{(B)} \circ \mathcal{S}_{t_{k+1}\leftarrow t_k}^{(A)} \Big\|_{\mathcal{L}(B)} \leq Me^\omega,$$

where $B$ is a Banach space of observables (e.g., equipped with the norm $\|\cdot\|_\infty + \|D\cdot\|_\infty$), $M \geq 1$, and $\omega \in \mathbb{R}$. This ensures that the operator splitting scheme remains stable over multiple time steps.

To ensure the stability of the Lie–Trotter splitting scheme, we need to bound the operator norm of the composition of the propagators $\mathcal{S}_{t_{k+1}\leftarrow t_k}^{(B)} \circ \mathcal{S}_{t_{k+1}\leftarrow t_k}^{(A)}$ over $n$ steps, where $\mathcal{S}^{(A)}$ and $\mathcal{S}^{(B)}$ are the solution operators (propagators) associated with the flows $\Phi^{(A)}$ and $\Phi^{(B)}$ acting on a Banach space of observables $B$. We define $B$ as the space of test functions $\varphi : \mathcal{U} \to \mathbb{R}$ equipped with the norm:

$$\|\varphi\|_B = \|\varphi\|_\infty + \|D\varphi\|_\infty,$$

where $\|\varphi\|_\infty = \sup_{u \in \mathcal{U}} |\varphi(u)|$ and $\|D\varphi\|_\infty = \sup_{u \in \mathcal{U}} \|D_u\varphi(u)\|_{\mathcal{U}^*}$.

*Proof.* We proceed by analyzing the action of the propagators $\mathcal{S}_{t+h\leftarrow t}^{(A)}$ and $\mathcal{S}_{t+h\leftarrow t}^{(B)}$ over a single time step $[t, t + h]$ with $h = \tau$. The propagator $\mathcal{S}_{t+h\leftarrow t}^{(A)}$ corresponds to the flow $\Phi_{t+h\leftarrow t}^{(A)}$, which evolves the field $f$ according to:

$$\frac{d}{ds}f(s) = v_f(s, f(s), g), \qquad g(s) = g(t), \quad s \in [t, t + h],$$

and similarly for $\mathcal{S}^{(B)}$. For a test function $\varphi \in B$, the propagator is defined as:

$$\mathcal{S}_{t+h\leftarrow t}^{(A)}\varphi(u) = \varphi(\Phi_{t+h\leftarrow t}^{(A)}(u)).$$

We need to bound the operator norm:

$$\|\mathcal{S}_{t+h\leftarrow t}^{(A)}\|_{\mathcal{L}(B)} = \sup_{\|\varphi\|_B \leq 1} \|\mathcal{S}_{t+h\leftarrow t}^{(A)}\varphi\|_B = \sup_{\|\varphi\|_B \leq 1} \left( \|\mathcal{S}_{t+h\leftarrow t}^{(A)}\varphi\|_\infty + \|D(\mathcal{S}_{t+h\leftarrow t}^{(A)}\varphi)\|_\infty \right).$$

### B.2.1 Bound on the sup-norm

Since $\Phi^{(A)}_{t+h\leftarrow t}$ is a flow map on $\mathcal{U}$, we have:

$$\|\mathcal{S}^{(A)}_{t+h\leftarrow t}\varphi\|_\infty = \sup_{u\in\mathcal{U}} |\varphi(\Phi^{(A)}_{t+h\leftarrow t}(u))| \leq \sup_{u'\in\mathcal{U}} |\varphi(u')| = \|\varphi\|_\infty,$$

because the flow map simply reparametrizes the domain of $\varphi$.

### B.2.2 Bound on the gradient norm

The Fréchet derivative of $\mathcal{S}^{(A)}_{t+h\leftarrow t}\varphi$ is:

$$D(\mathcal{S}^{(A)}_{t+h\leftarrow t}\varphi)(u) = D\varphi(\Phi^{(A)}_{t+h\leftarrow t}(u)) \cdot D\Phi^{(A)}_{t+h\leftarrow t}(u),$$

where $D\Phi^{(A)}_{t+h\leftarrow t}(u) : \mathcal{U} \to \mathcal{U}$ is the Jacobian of the flow map. We need to bound $\|D\Phi^{(A)}_{t+h\leftarrow t}(u)\|_{\mathcal{L}(\mathcal{U})}$. The flow $\Phi^{(A)}$ satisfies the ODE:

$$\frac{d}{ds}\Phi^{(A)}_{s\leftarrow t}(u) = v^{(A)}(s, \Phi^{(A)}_{s\leftarrow t}(u)), \quad \Phi^{(A)}_{t\leftarrow t}(u) = u.$$

The Jacobian $D\Phi^{(A)}_{s\leftarrow t}(u)$ evolves according to the variational equation:

$$\frac{d}{ds}D\Phi^{(A)}_{s\leftarrow t}(u) = Dv^{(A)}(s, \Phi^{(A)}_{s\leftarrow t}(u)) \cdot D\Phi^{(A)}_{s\leftarrow t}(u), \quad D\Phi^{(A)}_{t\leftarrow t}(u) = I.$$

By assumption (A1), $v^{(A)}(t, u) = (v_f(t, f, g), 0)$ satisfies the Lipschitz condition:

$$\|v^{(A)}(t, u) - v^{(A)}(t, \tilde{u})\|_{\mathcal{U}} = \|v_f(t, f, g) - v_f(t, \tilde{f}, \tilde{g})\|_{\mathcal{F}} \leq L_R\|u - \tilde{u}\|_{\mathcal{U}},$$

for $u, \tilde{u}$ in a ball of radius $R$. Thus, the operator norm of the Fréchet derivative $Dv^{(A)}(t, u)$ is bounded by $L_R$. Applying Grönwall's inequality to the variational equation:

$$\|D\Phi^{(A)}_{t+h\leftarrow t}(u)\|_{\mathcal{L}(\mathcal{U})} \leq e^{\int_t^{t+h} L_R\, ds} = e^{L_R h}.$$

Therefore:

$$\|D(\mathcal{S}^{(A)}_{t+h\leftarrow t}\varphi)\|_\infty = \sup_{u\in\mathcal{U}} \|D\varphi(\Phi^{(A)}_{t+h\leftarrow t}(u)) \cdot D\Phi^{(A)}_{t+h\leftarrow t}(u)\|_{\mathcal{U}^*} \leq \|D\varphi\|_\infty e^{L_R h}.$$

Combining both bounds:

$$\|\mathcal{S}^{(A)}_{t+h\leftarrow t}\varphi\|_B \leq \|\varphi\|_\infty + e^{L_R h}\|D\varphi\|_\infty \leq e^{L_R h}(\|\varphi\|_\infty + \|D\varphi\|_\infty) = e^{L_R h}\|\varphi\|_B,$$

so:

$$\|\mathcal{S}^{(A)}_{t+h\leftarrow t}\|_{\mathcal{L}(B)} \leq e^{L_R h}.$$

Similarly, for $\mathcal{S}^{(B)}_{t+h\leftarrow t}$, we obtain:

$$\|\mathcal{S}^{(B)}_{t+h\leftarrow t}\|_{\mathcal{L}(B)} \leq e^{L_R h},$$

since $v^{(B)}$ satisfies the same Lipschitz bound by (A1).

### B.2.3 Composition over one step

For the composition $\mathcal{S}^{(B)}_{t+h\leftarrow t} \circ \mathcal{S}^{(A)}_{t+h\leftarrow t}$:

$$\|\mathcal{S}^{(B)}_{t+h\leftarrow t} \circ \mathcal{S}^{(A)}_{t+h\leftarrow t}\|_{\mathcal{L}(B)} \leq \|\mathcal{S}^{(B)}_{t+h\leftarrow t}\|_{\mathcal{L}(B)} \cdot \|\mathcal{S}^{(A)}_{t+h\leftarrow t}\|_{\mathcal{L}(B)} \leq e^{L_R h} \cdot e^{L_R h} = e^{2L_R h}.$$

### B.2.4 COMPOSITION OVER $n$ STEPS

Over $n = 1/\tau$ steps covering $[0, 1]$, with $h = \tau = 1/n$, the total composition is:

$$\left\| \prod_{k=0}^{n-1} \mathcal{S}_{t_{k+1} \leftarrow t_k}^{(B)} \circ \mathcal{S}_{t_{k+1} \leftarrow t_k}^{(A)} \right\|_{\mathcal{L}(B)} \leq \prod_{k=0}^{n-1} e^{2L_R \tau} = e^{2L_R \cdot n \cdot \tau} = e^{2L_R \cdot 1} = e^{2L_R}.$$

Thus, we set $M = 1$ and $\omega = 2L_R$, satisfying the required bound:

$$\left\| \prod_{k=0}^{n-1} \mathcal{S}_{t_{k+1} \leftarrow t_k}^{(B)} \circ \mathcal{S}_{t_{k+1} \leftarrow t_k}^{(A)} \right\|_{\mathcal{L}(B)} \leq e^{2L_R}.$$

$\square$

### B.3 PROOF OF CONSISTENCY

Consistency means the local error (difference between the exact flow and the split flow on one subinterval) vanishes as $h \to 0$. For every $t \in [0, 1)$ and $\varphi \in \mathcal{T}$,

$$\lim_{h \to 0} \frac{\left( \mathcal{S}_{t+h \leftarrow t}^{(B)} \circ \mathcal{S}_{t+h \leftarrow t}^{(A)} \varphi - \varphi \right)}{h} = \mathcal{L}(t)\varphi$$

uniformly on bounded subsets of $\mathcal{U}$.

*Proof.* Fix a bounded set $K \subset \mathcal{U}$ and $u \in K$. We expand $\mathcal{S}^{(A)}\varphi(u) = \varphi(\Phi^{(A)}(u))$ using Taylor's theorem for Fréchet differentiable functions. For small $h$, the A-flow satisfies:

$$\Phi_{t+h \leftarrow t}^{(A)}(u) = u + hv^{(A)}(t, u) + R_A(h, u),$$

where the remainder $R_A(h, u)$ satisfies $\frac{\|R_A(h,u)\|}{h} \to 0$ as $h \to 0$ (uniformly on $K$, by local Lipschitzness of $v$).

Expanding $\varphi$ around $u$:

$$\varphi(\Phi^{(A)}(u)) = \varphi(u) + \langle D_u\varphi(u), hv^{(A)}(t, u) + R_A(h, u) \rangle + o(\|hv^{(A)} + R_A\|).$$

Dividing by $h$:

$$\frac{1}{h} \left( \mathcal{S}^{(A)}\varphi(u) - \varphi(u) \right) = \langle D_u\varphi(u), v^{(A)}(t, u) \rangle + \frac{1}{h}\langle D_u\varphi(u), R_A(h, u) \rangle + o\left( \frac{\|hv^{(A)} + R_A\|}{h} \right).$$

As $h \to 0$, the second and third terms vanish (uniformly on $K$), so:

$$\frac{1}{h} \left( \mathcal{S}^{(A)}\varphi(u) - \varphi(u) \right) \to \langle D_u\varphi(u), v^{(A)}(t, u) \rangle \quad \text{uniformly on } K.$$

Now apply the B-flow to $\mathcal{S}^{(A)}\varphi$. Expanding $\mathcal{S}^{(B)}\mathcal{S}^{(A)}\varphi(u) = \varphi(\Phi^{(B)}(\Phi^{(A)}(u)))$ similarly:

$$\Phi_{t+h \leftarrow t}^{(B)}(\Phi^{(A)}(u)) = \Phi^{(A)}(u) + hv^{(B)}(t, \Phi^{(A)}(u)) + R_B(h, u),$$

with $\frac{\|R_B(h,u)\|}{h} \to 0$ uniformly on $K$. Expanding $\varphi$ around $\Phi^{(A)}(u)$:

$$\varphi(\Phi^{(B)}(\Phi^{(A)}(u))) = \varphi(\Phi^{(A)}(u)) + h\langle D_u\varphi(\Phi^{(A)}(u)), v^{(B)}(t, \Phi^{(A)}(u)) \rangle + o(h).$$

Substituting the A-step expansion:

$$\mathcal{S}^{(B)}\mathcal{S}^{(A)}\varphi(u) = \varphi(u) + h\langle D_u\varphi(u), v^{(A)}(t, u) \rangle + h\langle D_u\varphi(u), v^{(B)}(t, u) \rangle + o(h),$$

since $D_u\varphi(\Phi^{(A)}(u)) = D_u\varphi(u) + o(1)$ (continuity of $D_u\varphi$) and $v^{(B)}(t, \Phi^{(A)}(u)) = v^{(B)}(t, u) + o(1)$ (continuity of $v$).

Combining terms and dividing by $h$:

$$\frac{1}{h} \left( \mathcal{S}^{(B)}\mathcal{S}^{(A)}\varphi(u) - \varphi(u) \right) = \langle D_u\varphi(u), v(t, u) \rangle + o(1) = -\mathcal{L}(t)\varphi(u) + o(1).$$

Taking $h \to 0$ gives the result. $\square$

## B.4 PROOF OF CONVERGENCE

Below, we provide a complete and rigorous proof of the convergence result stated in the paragraph. The proof is constructed based on standard results from operator splitting theory for ODEs in Hilbert spaces, error estimates for Lie-Trotter methods, and bounds on Wasserstein distances for pushforward measures. It assumes the conditions from Assumptions (A1)–(A3) as defined in the document. The proof is divided into steps for clarity: first, bounding the splitting error with exact vector fields, then incorporating the learning error, and finally combining them via the triangle inequality.

### B.4.1 PRELIMINARIES

Recall that the coupled evolution is governed by the flow map $\Phi_t : \mathcal{U} \to \mathcal{U}$, satisfying the ODE

$$\frac{d}{dt}\Phi_t(u) = v(t, \Phi_t(u)), \quad \Phi_0(u) = u, \quad u = (f, g) \in \mathcal{U},$$

where $v(t, u) = (V_f(t, f, g), V_g(t, f, g))$ is the velocity field from the weak continuity equation Eq. 2. The true measure at time 1 is $\mu_1 = \Phi_1 \# \mu_0$ (pushforward of $\mu_0$ under $\Phi_1$).

The Lie-Trotter splitting approximates this flow with component flows $\Phi_h^{(A)}$ and $\Phi_h^{(B)}$, where: - For $\Phi_t^{(A)}$: $\frac{d}{dt}\Phi_t^{(A)}(u) = v^{(A)}(t, \Phi_t^{(A)}(u))$ with $v^{(A)}(t, u) = (V_f(t, f, g), 0)$, - For $\Phi_t^{(B)}$: $\frac{d}{dt}\Phi_t^{(B)}(u) = v^{(B)}(t, \Phi_t^{(B)}(u))$ with $v^{(B)}(t, u) = (0, V_g(t, f, g))$.

The split flow over one time step $h$ is $\Phi_h^{(split)} = \Phi_h^{(B)} \circ \Phi_h^{(A)}$. Over the interval $[0, 1]$ with $N = 1/\tau$ (number of steps) and step size $h = \tau$, the approximate measure from exact splitting is $\mu_1^{(\tau)} = \left(\Phi_\tau^{(split)}\right)^N \# \mu_0$ (pushforward under $N$-fold composition of $\Phi_\tau^{(split)}$).

For learned velocity fields $\hat{v}_f$ and $\hat{v}_g$, the learned component flows are $\hat{\Phi}_h^{(A)}$ (governed by $\hat{v}^{(A)} = (\hat{v}_f, 0)$) and $\hat{\Phi}_h^{(B)}$ (governed by $\hat{v}^{(B)} = (0, \hat{v}_g)$). The learned split flow is $\hat{\Phi}_h^{(split)} = \hat{\Phi}_h^{(B)} \circ \hat{\Phi}_h^{(A)}$, and the corresponding learned approximate measure is $\mu_1^{(\tau,\text{learn})} = \left(\hat{\Phi}_\tau^{(split)}\right)^N \# \mu_0$.

We assume $\mu_0 \in \mathcal{P}_2(\mathcal{U})$ (probability measures on $\mathcal{U}$ with finite second moment) and use the Wasserstein-1 distance:

$$W_1(\mu, \nu) = \sup_{\|\phi\|_{\text{Lip}} \leq 1} \left| \int_{\mathcal{U}} \phi \, d\mu - \int_{\mathcal{U}} \phi \, d\nu \right|,$$

where $\|\phi\|_{\text{Lip}} = \sup_{\substack{u, \tilde{u} \in \mathcal{U} \\ u \neq \tilde{u}}} \frac{|\phi(u) - \phi(\tilde{u})|}{\|u - \tilde{u}\|_{\mathcal{U}}}$ denotes the Lipschitz constant of $\phi$.

### B.4.2 STEP 1: LOCAL SPLITTING ERROR WITH EXACT FIELDS

Under Assumption (A1) (local Lipschitz continuity and polynomial growth of $v$), the flows $\Phi_t$, $\Phi_t^{(A)}$, and $\Phi_t^{(B)}$ exist and are unique for all $t \in [0, 1]$ (Picard-Lindelöf theorem for ODEs in Banach spaces).

To analyze the local error (error over one step $h$), we use Taylor expansions of the flows around $u$: - For the true flow: $\Phi_h(u) = u + hv(t, u) + \frac{h^2}{2}\frac{d}{dt}v(t, u) + O(h^3)$, - For the $A$-component flow: $\Phi_h^{(A)}(u) = u + hv^{(A)}(t, u) + \frac{h^2}{2}\frac{d}{dt}v^{(A)}(t, u) + O(h^3)$, - For the $B$-component flow: $\Phi_h^{(B)}(\tilde{u}) = \tilde{u} + hv^{(B)}(t, \tilde{u}) + \frac{h^2}{2}\frac{d}{dt}v^{(B)}(t, \tilde{u}) + O(h^3)$ (substitute $\tilde{u} = \Phi_h^{(A)}(u)$).

Substituting $\Phi_h^{(A)}(u)$ into $\Phi_h^{(B)}$, the split flow expands to:

$$\Phi_h^{(split)}(u) = u + h\left(v^{(A)} + v^{(B)}\right) + \frac{h^2}{2}\left(\frac{d}{dt}v^{(A)} + \frac{d}{dt}v^{(B)} + [v^{(A)}, v^{(B)}]\right) + O(h^3),$$

where $[v^{(A)}, v^{(B)}] = Dv^{(B)} \cdot v^{(A)} - Dv^{(A)} \cdot v^{(B)}$ is the Lie bracket (commutator) of $v^{(A)}$ and $v^{(B)}$ (with $Dv$ denoting the Fréchet derivative of $v$).

Since $v = v^{(A)} + v^{(B)}$ and $\frac{d}{dt}v = \frac{d}{dt}v^{(A)} + \frac{d}{dt}v^{(B)}$ (component-wise additivity), the local error of the split flow is:

$$\Phi_h(u) - \Phi_h^{(split)}(u) = \frac{h^2}{2}[v^{(A)}, v^{(B)}](t, u) + O(h^3).$$

Under Assumption (A1), the Lie bracket $[v^{(A)}, v^{(B)}]$ is bounded by a constant $K_R > 0$ on any ball $B_R = \{u \in \mathcal{U} \mid \|u\|_{\mathcal{U}} \leq R\}$ (local boundedness follows from local Lipschitz continuity of $v$). Thus, the local error in the $\|\cdot\|_{\mathcal{U}}$-norm satisfies:

$$\|\Phi_h(u) - \Phi_h^{(split)}(u)\|_{\mathcal{U}} = O(h^2).$$

For observable propagators (acting on test functions $\varphi$), define $\mathcal{S}_h^{(A)}\varphi(u) = \varphi(\Phi_h^{(A)}(u))$ and $\mathcal{S}_h^{(B)}\varphi(u) = \varphi(\Phi_h^{(B)}(u))$. The local splitting error for propagators is:

$$\mathcal{R}_h(\varphi; t, u) = h^{-1}\left(\mathcal{S}_{t+h \leftarrow t}^{(B)} \circ \mathcal{S}_{t+h \leftarrow t}^{(A)}\varphi(u) - \varphi(u)\right) - \mathcal{L}(t)\varphi(u) = O(h),$$

where $\mathcal{L}(t) = v(t, \cdot) \cdot D$ is the generator of the true flow. This holds for cylindrical test functions $\varphi \in \mathcal{T}$ with bounded Fréchet derivatives (Assumption (A2) on well-posedness of the continuity equation).

### B.4.3   Step 2: Global Splitting Error with Exact Fields

To extend the local error to the global error over $[0, 1]$, consider the $N$-fold composition of the split flow: $\left(\Phi_\tau^{(split)}\right)^N = \Phi_\tau^{(split)} \circ \Phi_\tau^{(split)} \circ \cdots \circ \Phi_\tau^{(split)}$ (N times), with $N\tau = 1$.

By the Lady Windermere's Fan Lemma (a standard tool for operator splitting(Hairer et al., 2006)) or a telescoping sum argument for splitting methods, the global error of the flow map satisfies:

$$\|\Phi_1(u) - \left(\Phi_\tau^{(split)}\right)^N(u)\|_{\mathcal{U}} \leq Ce^L\tau,$$

where: - $C > 0$ depends on the commutator bound $K_R$ (from Step 1) and the time horizon $T = 1$, - $L > 0$ is the local Lipschitz constant of $v$ (from Assumption (A1)).

The bound assumes trajectories stay in a bounded ball $B_R$ (justified by the polynomial growth of $v$ and finite second moment of $\mu_0$, which prevents trajectories from escaping to infinity).

To translate this flow error to a measure error (Wasserstein-1 distance), use the definition of $W_1$ and Lipschitz continuity of test functions:

$$W_1(\mu_1, \mu_1^{(\tau)}) = W_1(\Phi_1 \# \mu_0, \left(\Phi_\tau^{(split)}\right)^N \# \mu_0) \leq \sup_{\|\phi\|_{\text{Lip}} \leq 1} \int_{\mathcal{U}} \left|\phi(\Phi_1(u)) - \phi\left(\left(\Phi_\tau^{(split)}\right)^N(u)\right)\right| d\mu_0(u).$$

By Lipschitz continuity of $\phi$, $|\phi(\Phi_1(u)) - \phi\left(\left(\Phi_\tau^{(split)}\right)^N(u)\right)| \leq \|\Phi_1(u) - \left(\Phi_\tau^{(split)}\right)^N(u)\|_{\mathcal{U}}$.
Taking the supremum over $\phi$ and using the flow error bound:

$$W_1(\mu_1, \mu_1^{(\tau)}) \leq \int_{\mathcal{U}} \|\Phi_1(u) - \left(\Phi_\tau^{(split)}\right)^N(u)\|_{\mathcal{U}} d\mu_0(u) \leq C_{\text{stab}}\tau,$$

where $C_{\text{stab}} = Ce^L$ incorporates stability constants (from Section 3.3) and the time horizon. This is a Chernoff-type convergence result in the weak sense, extended to $W_1$ via Lipschitz bounds.

### B.4.4   Step 3: Learning Error in Flows

Under Assumption (A3) (accuracy of learned velocity fields), the learned fields $\hat{v}_f, \hat{v}_g$ satisfy uniform bounds on bounded sets $B_R$:

$$\|V_f(t, u) - \hat{v}_f(t, u)\|_{\mathcal{F}} \leq \varepsilon_f, \quad \|V_g(t, u) - \hat{v}_g(t, u)\|_{\mathcal{G}} \leq \varepsilon_g,$$

for all $t \in [0, 1]$ and $u \in B_R$. Define the total learning error as $\varepsilon_{\text{tot}} = \varepsilon_f + \varepsilon_g$, so:

$$\|v(t, u) - \hat{v}(t, u)\|_{\mathcal{U}} \leq \varepsilon_{\text{tot}} \quad \forall t \in [0, 1], u \in B_R.$$

Let $\hat{\Phi}_t$ denote the learned flow (governed by $\hat{v}$): $\frac{d}{dt}\hat{\Phi}_t(u) = \hat{v}(t, \hat{\Phi}_t(u))$ with $\hat{\Phi}_0(u) = u$. Define the flow error $e_t(u) = \Phi_t(u) - \hat{\Phi}_t(u)$; its time derivative satisfies:

$$\frac{d}{dt}e_t(u) = v(t, \Phi_t(u)) - \hat{v}(t, \hat{\Phi}_t(u)) = \underbrace{v(t, \Phi_t(u)) - v(t, \hat{\Phi}_t(u))}_{T_1} + \underbrace{v(t, \hat{\Phi}_t(u)) - \hat{v}(t, \hat{\Phi}_t(u))}_{T_2}.$$

Bounding each term: - $T_1$: By local Lipschitz continuity of $v$ (Assumption (A1)), $\|T_1\|_{\mathcal{U}} \leq L_R\|e_t(u)\|_{\mathcal{U}}$, - $T_2$: By Assumption (A3), $\|T_2\|_{\mathcal{U}} \leq \varepsilon_{\text{tot}}$.

Combining these, the error ODE becomes:

$$\frac{d}{dt}\|e_t(u)\|_{\mathcal{U}} \leq L_R\|e_t(u)\|_{\mathcal{U}} + \varepsilon_{\text{tot}}.$$

Applying Gronwall's Inequality (with initial condition $\|e_0(u)\|_{\mathcal{U}} = 0$):

$$\|e_t(u)\|_{\mathcal{U}} \leq \varepsilon_{\text{tot}} t e^{L_R t}.$$

For $t = 1$ (final time), this simplifies to:

$$\|\Phi_1(u) - \hat{\Phi}_1(u)\|_{\mathcal{U}} \leq \varepsilon_{\text{tot}} e^{L_R}.$$

Translating to a measure error (via $W_1$), similar to Step 2:

$$W_1(\Phi_1 \# \mu_0, \hat{\Phi}_1 \# \mu_0) \leq \varepsilon_{\text{tot}} e^{L_R}.$$

### B.4.5 STEP 4: TOTAL ERROR FOR LEARNED SPLITTING

We now analyze the total error of the learned splitting scheme, where the learned approximate measure is $\mu_1^{(\tau,\text{learn})} = \left(\hat{\Phi}_\tau^{(split)}\right)^N \# \mu_0$. Here, $\hat{\Phi}_\tau^{(split)} = \hat{\Phi}_\tau^{(B)} \circ \hat{\Phi}_\tau^{(A)}$ denotes the learned split flow: $\hat{\Phi}_\tau^{(A)}$ evolves under the learned field $\hat{v}^{(A)} = (\hat{v}_f, 0)$, and $\hat{\Phi}_\tau^{(B)}$ evolves under $\hat{v}^{(B)} = (0, \hat{v}_g)$.

First, we bound the splitting error of the learned flow (analogous to Step 2 for exact fields). Under Assumption (A3) (learned fields are locally Lipschitz with constant $L_{\text{model}} \approx L_R$, matching the Lipschitz regularity of the true fields), the key properties of the split flow are preserved:

- The Lie bracket of the learned components $[\hat{v}^{(A)}, \hat{v}^{(B)}]$ is bounded on bounded sets (by $K_R' \approx K_R$, since Lipschitz continuity implies bounded Fréchet derivatives).

- The global splitting error for the learned flow $\left(\hat{\Phi}_\tau^{(split)}\right)^N$ (relative to the learned full flow $\hat{\Phi}_1$) is $O(\tau)$, with stability constant $\hat{C}_{\text{stab}} \approx C_{\text{stab}}$ (the smallness of $\varepsilon_{\text{tot}}$ ensures $L_{\text{model}}$ and $K_R'$ remain close to $L_R$ and $K_R$).

Next, we bound the error between the exact split measure $\mu_1^{(\tau)}$ and the learned split measure $\mu_1^{(\tau,\text{learn})}$. This error arises from the discrepancy between the exact split flow $\left(\Phi_\tau^{(split)}\right)^N$ and the learned split flow $\left(\hat{\Phi}_\tau^{(split)}\right)^N$. Using an argument analogous to Step 3 (Gronwall's inequality for split flows): - For each step $k \in \{0, 1, \ldots, N-1\}$, the error between $\Phi_\tau^{(split)}$ and $\hat{\Phi}_\tau^{(split)}$ is bounded by $O(\varepsilon_{\text{tot}}\tau)$ (local error), - Compounding this over $N = 1/\tau$ steps gives a global flow error of $O(\varepsilon_{\text{tot}})$, with constant $\hat{C}$ (depending on $L_{\text{model}}$ and $T = 1$).

Translating this to the Wasserstein-1 distance (via Lipschitz continuity of test functions and finite moments of $\mu_0$):

$$W_1\left(\mu_1^{(\tau)}, \mu_1^{(\tau,\text{learn})}\right) \leq \hat{C}\varepsilon_{\text{tot}}.$$

Finally, we combine the two error components using the triangle inequality for the Wasserstein-1 distance:

$$W_1\left(\mu_1^{(\tau,\text{learn})}, \mu_1\right) \leq W_1\left(\mu_1^{(\tau,\text{learn})}, \mu_1^{(\tau)}\right) + W_1\left(\mu_1^{(\tau)}, \mu_1\right).$$

Substituting the bounds from Step 2 ($W_1(\mu_1^{(\tau)}, \mu_1) \leq C_{\text{stab}}\tau$) and the above learned split error:

$$W_1\left(\mu_1^{(\tau,\text{learn})}, \mu_1\right) \leq \hat{C}\varepsilon_{\text{tot}} + C_{\text{stab}}\tau.$$

Defining $C_{\text{tot}} = \max\{C_{\text{stab}}, \hat{C}\}$ (a constant depending on the Lipschitz constants $L_R, L_{\text{model}}$, commutator bounds $K_R, K'_R$, stability constants $M, \omega$ from Assumption (A3), and time horizon $T = 1$), this simplifies to:

$$W_1\left(\mu_1^{(\tau,\text{learn})}, \mu_1\right) \leq C_{\text{tot}}\left(\tau + \varepsilon_{\text{tot}}\right).$$

This completes the convergence proof. A key implicit assumption—trajectories remain in a bounded set $B_R$ where the uniform Lipschitz and learning error bounds apply—is justified by: 1. The polynomial growth condition on $v$ (Assumption (A1)), which prevents trajectories from escaping to infinity in finite time, 2. The finite second moment of $\mu_0$ (hence finite first moment), which ensures the bulk of trajectories stay within $B_R$ for some $R > 0$.

For infinite-dimensional spaces $\mathcal{U}$ (e.g., function spaces), the weak formulation of the continuity equation (Assumption (A2)) ensures that all integrals and pushforward measures are well-defined (measurable and integrable), as cylindrical test functions avoid issues with infinite-dimensional norms.

## C  EXPLANATIONS IN THE FFM FRAMEWORK

### C.1  OVERVIEW OF FUNCTIONAL FLOW MATCHING

Functional Flow Matching (FFM) is a function-space generative model that generalizes the recently-introduced Flow Matching model to operate in infinite-dimensional spaces (Kerrigan et al., 2023). The key motivation for developing models in functional spaces stems from the fact that many real-world data types, such as time series, solutions to partial differential equations (PDEs), and audio signals, are inherently infinite-dimensional and naturally represented as functions. Traditional generative models that operate on finite-dimensional discretizations (e.g., grids) are often tied to specific resolutions, making them ill-posed in the limit of zero discretization error and difficult to transfer across different discretizations. In contrast, functional approaches like FFM are discretization-invariant, allowing generation of functions that can be evaluated at any arbitrary points, ensuring consistency and flexibility for infinite-dimensional data.

FFM works by defining a path of probability measures that interpolates between a fixed Gaussian measure and the data distribution, then learning a vector field on the underlying space of functions that generates this path. Unlike density-based methods, FFM is purely measure-theoretic, avoiding issues with non-existent densities in infinite dimensions (e.g., no analogue of Lebesgue measure exists in Banach spaces). It enables simulation-free training via a regression objective, making it efficient and suitable for function spaces.

Let $X \subset \mathbb{R}^d$ and consider a real separable Hilbert space $\mathcal{F}$ of functions $f : X \to \mathbb{R}$ equipped with the Borel $\sigma$-algebra $\mathcal{B}(\mathcal{F})$. The goal is to sample from a data distribution $\nu$ on $\mathcal{F}$. FFM constructs a path of measures $(\mu_t)_{t\in[0,1]}$, where $\mu_0$ is a fixed reference measure (e.g., Gaussian) and $\mu_1 \approx \nu$.

A time-dependent vector field $v : [0, 1] \times \mathcal{F} \to \mathcal{F}$ generates the flow $\phi : [0, 1] \times \mathcal{F} \to \mathcal{F}$ via the ODE:

$$\partial_t \phi_t(g) = v_t(\phi_t(g)), \quad \phi_0(g) = g.$$

The path is $\mu_t = [\phi_t]_\# \mu_0$, the pushforward of $\mu_0$.

### C.2  MATHEMATICAL JUSTIFICATION: MARGINAL-TO-JOINT CONSTRUCTION AND SPLITTING

We recall the marginalization identity established in functional flow matching (Theorem 1 of Kerrigan et al. (2023)): if for each $f$ the conditional path $\mu_t^f$ is generated by a vector field $v_t^f$ and is absolutely continuous with respect to the marginal $\mu_t$, then the averaged vector field

$$\bar{v}_t(u) = \int v_t^f(u) \frac{d\mu_t^f}{d\mu_t}(u)\, d\nu^f(f)$$

generates the marginal path $\mu_t = \int \mu_t^f \, d\nu^f(f)$, and the pair $(\bar{v}_t, \mu_t)$ satisfies the weak continuity equation.

In our split setting, we apply this construction coordinate-wise. For almost every joint sample $(f, g)$ from $\nu$, let $v_t^{(f|g)}$ be the conditional velocity for $f$ with $g$ frozen, and let $w_t^{(g|f)}$ be the conditional velocity for $g$ with $f$ frozen. Lifting these to the product space by zero-padding, we define

$$v_t^{(A)}(f, g) := \Big( \int v_t^{(f'|g)}(f) \, \frac{d\mu_t^{f'}(f)}{d\mu_t(f)} \, d\nu^f(f'), \, 0 \Big),$$

and analogously

$$v_t^{(B)}(f, g) := \Big( 0, \, \int w_t^{(g'|f)}(g) \, \frac{d\mu_t^{g'}(g)}{d\mu_t(g)} \, d\nu^g(g') \Big).$$

Under absolute continuity and integrability, these define the A- and B- component fields, yielding generators

$$\mathcal{L}_A(t) = \langle v_t^{(A)}, D_u \cdot \rangle, \qquad \mathcal{L}_B(t) = \langle v_t^{(B)}, D_u \cdot \rangle,$$

and hence the full Liouville operator $\mathcal{L} = \mathcal{L}_A + \mathcal{L}_B$. This provides a principled justification for learning $\hat{v}_t^f$ and $\hat{w}_t^g$ separately and composing their lifted actions in the joint space.

### C.3 Equivalence to paired-training schemes via bijective reparametrizations

Several applied works, such as the marginal-to-PDE framework of Zhang et al. (2025), train conditional models with mixed parameterizations. For example, one may train $v_t^f$ using pairs $(f_t, g_1)$ and $v_t^g$ using $(f_1, g_t)$, where $f_1, g_1$ denote prescribed boundary or terminal states. Despite appearing different, these paired schemes are mathematically equivalent to the decoupled training described above, once we account for bijective reparametrizations between the complementary coordinates (e.g., mapping $g_t$ to $g_1$ along the frozen B-flow).

**Theorem C.1** (Equivalence under bijective coordinate reparametrization.). *Let $\mathcal{R}_g : \mathcal{G} \to \mathcal{G}$ be a measurable bijection such that for each frozen $f$, the reparametrization between the complementary coordinates in the two training schemes is given by $\mathcal{R}_g$ (and analogously $\mathcal{R}_f$ for the $f$-coordinate). Assume $\mathcal{R}_g, \mathcal{R}_f$ and their inverses are Lipschitz on bounded supports and preserve conditional path measures. Then training $v_t^f$ with pairs $(f_t, g_1)$ is equivalent, up to reparametrization, to training with pairs $(f_t, g_t)$; similarly for $v_t^g$. Consequently, inference using $\hat{v}_t^f, \hat{w}_t^g$ trained under either scheme yields the same joint pushforward after mapping representations appropriately.*

*Proof.* Let $\Psi_g$ denote the representation map from scheme (A) to scheme (B), e.g. $\Psi_g(g_t) = g_1$. By assumption, $\Psi_g$ is bijective with Lipschitz inverse and pushes forward conditional measures correctly. For scheme (1) (pairs $(f_t, g_t)$), the training loss is

$$\mathcal{J}_1(\theta_f) = \mathbb{E}_f \, \mathbb{E}_{g \sim \mu_t^f} \big[ \|v_t^f(g) - u_t^f(g; \theta_f)\|^2 \big].$$

Changing variables $h = \Psi_g(g)$ gives

$$\mathcal{J}_1(\theta_f) = \mathbb{E}_f \, \mathbb{E}_{h \sim \Psi_g \# \mu_t^f} \big[ \|v_t^f(\Psi_g^{-1}(h)) - u_t^f(\Psi_g^{-1}(h); \theta_f)\|^2 \big].$$

Define $\tilde{v}_t^f(h) := v_t^f(\Psi_g^{-1}(h))$. Then this loss coincides with the scheme (2) objective (pairs $(f_t, g_1 = h)$) with target $\tilde{v}_t^f$. Thus the two schemes are equivalent up to pullback by $\Psi_g$. The same argument applies to $v_t^g$. At inference time, pushforwards commute with coordinate changes, so the joint pushforward is identical under either parametrization. $\qquad \square$

## D   Additional Details for dataset

**Synthetical dataset.** The "Easy distribution" is defined as a two-component Gaussian mixture. One component (weight 0.6) is centered at the specified mean, while the other (weight 0.4) is slightly shifted. The covariance matrices capture different correlation structures, with one showing a positive correlation of 0.8 and the other a negative correlation of -0.42, along with distinct variances across

dimensions. The "Complex distribution" is constructed from multiple equally weighted Gaussian components arranged along a circle. These components feature angle-dependent correlations, varying standard deviations, and a small isotropic Gaussian perturbation that enhances multimodality.

To construct datasets that preserve conditional information, we avoid direct joint sampling and instead generate two complementary datasets: one focusing on $p(y|x)$ and the other on $p(x|y)$. A portion of samples is drawn directly from the mixture to preserve global structure. For the $p(y|x)$ dataset, $x$ is drawn uniformly from [-3, 3], and $y$ is sampled from the conditional distribution $p(y|x)$. Conversely, for the $p(x|y)$ dataset, $y$ is drawn from a scaled Beta distribution over [-2, 2], and $x$ is sampled from $p(x|y)$. To further perturb the marginals, we introduce a mixing parameter that blends the original $x$ marginal with a uniform distribution and the original $y$ marginal with a scaled Beta distribution, while keeping the conditionals intact. The conditionals for Gaussian distributions are computed exactly and those for Gaussian mixtures are obtained via mixture-conditioning.

**Turek-Hron.** This simulation models a FSI scenario using the CFD solver (Lilypad (Weymouth & Yue, 2011)) where a flexible Bernoulli beam is placed downstream of a circular cylinder in a two-dimensional viscous flow. The system features a Reynolds number of Re=1000. Let us make $D$ denotes the diameter of cylinder. The spatially computational domain is $20D \times 14D$. The flexible beam has a length of $2D$ and thickness of $0.06D$, characterized by an elastic modulus of approximately 336,000 and density of 10. The simulation captures the coupled dynamics between the fluid flow and structural deformation over a 10 time units episode with a fixed timestep of 0.001. Key outputs include velocity fields, pressure distributions, and beam displacement data saved at each time step for experiments.

Normally, the original solver addresses a FSI problem, with one FSI trajectory computed as ground truth. To generate decoupled data, we implemented two solver modifications. First, for solid-to-fluid unidirectional coupling, we compute flow fields using only a single square cylinder (the side length is $D$) flow, then inversely calculate beam forces from this fluid data to control the beam, and finally compute resulting flow fields induced by its motion. Second, for fluid-to-solid unidirectional coupling, we generate flow fields using a $0.5D$ cylinder at each timestep to inversely compute beam positions. Both datasets contain no bidirectional coupling data, and crucially use flow conditions (square cylinder and reduced-diameter cylinder) completely distinct from the test set to prevent information leakage.

This scenario has extensive applications across diverse fields involving flexible structures in fluid environments. The methodology proves particularly valuable in aquatic ecosystem research, where it accurately models the dynamic behavior of submerged vegetation such as kelp forests and seagrass beds under varying current conditions, enabling scientists to understand their role in coastal protection and marine biodiversity conservation. In biomedical engineering, the framework facilitates critical hemodynamics studies by simulating blood flow interactions with deformable arterial walls, helping predict aneurysm rupture risks and optimize cardiovascular device designs like stents and artificial heart valves. The renewable energy sector benefits from applications in wave energy harvesting, where flexible oscillating elements mimic the motion of aquatic plants to efficiently capture ocean wave energy.

**Double Cylinder.** This simulation models a dual-cylinder fluid-structure interaction system where a fixed upstream cylinder is positioned ahead of a freely oscillating downstream cylinder that can vibrate in the y-direction. The flow is characterized by a Reynolds number Re=1000. Let us also make the diameter of cylinder $D$. The computational domain spans $8D \times 8D$ lengths ($128 \times 128$ grid points). Both cylinders have the same diameter $D$. The fixed upstream cylinder is positioned at $(1.5D, 4D)$ while the oscillating downstream cylinder is located at $(4D, 4D)$, creating a center-to-center spacing of $2.5D$. The downstream cylinder exhibits a mass ratio is 15.0 (relative to displaced fluid mass), dimensionless natural frequency is 0.01, and damping ratio is 0.8. The simulation employs a time step 0.1 over a total duration of 1000 time units, capturing characteristic vortex-induced vibration phenomena. Our method of generating decoupled data is similar to the Turek-Hron.

This scenario finds critical applications in underwater structural stability and multi-body hydrodynamic systems across various engineering domains. In offshore engineering, the model accurately predicts the complex wake interference effects between multiple platform columns subjected to

ocean currents, where the upstream cylinder's vortex shedding can trigger vortex-induced vibrations in the downstream cylinder, leading to structural fatigue. Additionally, the simulation supports the design of underwater cable systems with multi-point anchoring configurations, deep-sea aquaculture net cage arrays where inter-structure spacing affects fish habitat stability, tidal energy converter farms requiring optimal turbine spacing to balance energy harvesting efficiency with structural safety, and autonomous underwater vehicle swarms where wake interference affects formation stability and energy consumption.

It is worth noting that in our FSI experiments, when generating the structure-condition-on-fluid decoupled data, the prescribed fluid fields had strong periodicity, resulting in negligible structural deformation. This led to a limitation: the model never observed meaningful structural motion during training, and thus could not reasonably be expected to predict structural dynamics. To address this, in all experiments we replaced the structure-condition-on-fluid decoupled data with fully coupled data, while still using the fluid-condition-on-structure decoupled data (constructed as described above). Importantly, this "half-decoupled" setting is a fair compromise that does not undermine the core functionality of our method nor diminish the demonstration of its key advantages.

**Nculear Thermal Coupling.** The NT Coupling dataset is generated using the open-source MOOSE (Multiphysics Object-Oriented Simulation Environment) framework (Icenhour et al., 2018). The governing equations for the coupled evolution of the neutron flux ($\phi$), fuel temperature ($T_s$), and fluid temperature, velocity, and pressure ($T_f, \vec{u}, p$) are provided in the following equations. The physical fields are discretized on separate meshes: $64 \times 8$ for the fuel temperature, $64 \times 12$ for the fluid variables, and $64 \times 20$ for the neutron field. Neutron physics and fuel temperature are solved using the Finite Element Method (FEM), while the fluid domain is computed using the Finite Volume Method (FVM). All fields are subsequently interpolated onto a common grid to ensure spatial alignment. An adaptive time-stepping scheme is employed, and 16 output frames are recorded to capture the transient evolution of the system.

$$
\begin{cases}
\dfrac{1}{v}\dfrac{\partial \phi(x,y,t)}{\partial t} = D\Delta\phi + \big(v\Sigma_f - \Sigma_a(T)\big)\phi, \quad x \in [0, L_s + L_f],\ y \in [0, L_y],\ t \in [0,5], \\[2mm]
\qquad\qquad\qquad \phi(0,y,t) = f(y,t), \\[2mm]
\qquad \phi(L_s + L_f, y, t) = \phi(x,0,t) = \phi(x, L_y, t) = 0.
\end{cases}
\tag{6}
$$

$$
\begin{cases}
\rho c_p \dfrac{\partial T_s(x,y,t)}{\partial t} = \nabla \cdot (k_s \nabla T_s) + A\phi_s, \quad x \in [0, L_s],\ y \in [0, L_y],\ t \in [0,5], \\[2mm]
\qquad\qquad \dfrac{\partial T_s}{\partial y}(x,0,t) = \dfrac{\partial T_s}{\partial y}(x, L_y, t) = 0, \\[2mm]
\qquad\qquad T_s(L_s, y, t) = T_f(L_s, y, t).
\end{cases}
\tag{7}
$$

$$
\begin{cases}
\qquad \nabla \cdot \vec{u} = 0, \quad x \in [L_s, L_s + L_f],\ y \in [0, L_y],\ t \in [0,5], \\[2mm]
\rho\left(\dfrac{\partial \vec{u}}{\partial t} + \vec{u}\cdot\nabla\vec{u}\right) = -\nabla p + \mu\nabla^2\vec{u} + \vec{f}, \quad x \in [L_s, L_s + L_f],\ y \in [0, L_y],\ t \in [0,5], \\[2mm]
\rho c_p\left(\dfrac{\partial T_f}{\partial t} + \vec{u}\cdot\nabla T_f\right) = k_f\nabla^2 T_f, \quad x \in [L_s, L_s + L_f],\ y \in [0, L_y],\ t \in [0,5], \\[2mm]
\qquad\quad k_f\dfrac{\partial T_f}{\partial x}(L_s, y, t) = k_s\dfrac{\partial T_s}{\partial x}(L_s, y, t).
\end{cases}
\tag{8}
$$

We generate the data following the pre-iteration method (Zhang et al., 2025), ensuring the resulting dataset preserves the intrinsic multiphysics interactions. The sequence begins by holding the fluid variables and fuel temperature constant while solving for the neutron field. The resulting neutron distribution, together with the assumed constant fluid temperature, is then used to update the fuel

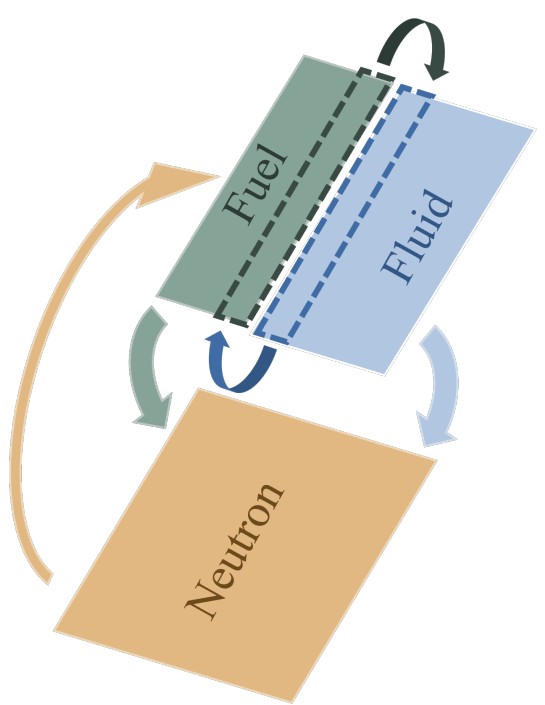

Figure 4: The geometric configuration and coupling mechanism of NT Coupling setting.

temperature field. The process continues by computing the fluid fields using the partially coupled states, followed by a final fully coupled pass over the neutron and fuel temperature equations. For simplicity in the coupling process, we use spatially distributed fuel temperature fields as inputs in place of the explicit heat flux.

## E  ADDITIONAL VISUALIZATION RESULTS

In this section, we present additional visualizations that were omitted from the main paper due to space limitations. Figure 5 shows the coupled simulation results for the same sample as in Figure 3, but using CNO as the backbone. Figure 6 provides analogous visualizations using FNO* as the backbone, on a sample from the Double-Cylinder dataset. In both cases—regardless of backbone or dataset, our proposed *GenCP* consistently delivers visually superior results compared to the baselines.

Furthermore, to illustrate the difference between decoupled data used for training and coupled data used as inferring, Figure 7 reports predictions of four physical fields under three settings: (i) training and inference both on decoupled data, (ii) training on decoupled data but inferring on coupled data using *GenCP*, and (iii) joint training and inference on coupled data. The results reveal that the coupled fields differ significantly in pattern from the decoupled ones, especially near structures. Nevertheless, our *GenCP* successfully learns the underlying physical interactions from decoupled data and incorporates them into the flow-matching generative process, thereby producing predictions that closely approximate the truly coupled solutions.

Figure 8 presents the visualized results of coupled inference with decoupled learning, using different paradigms on NT Coupling setting.

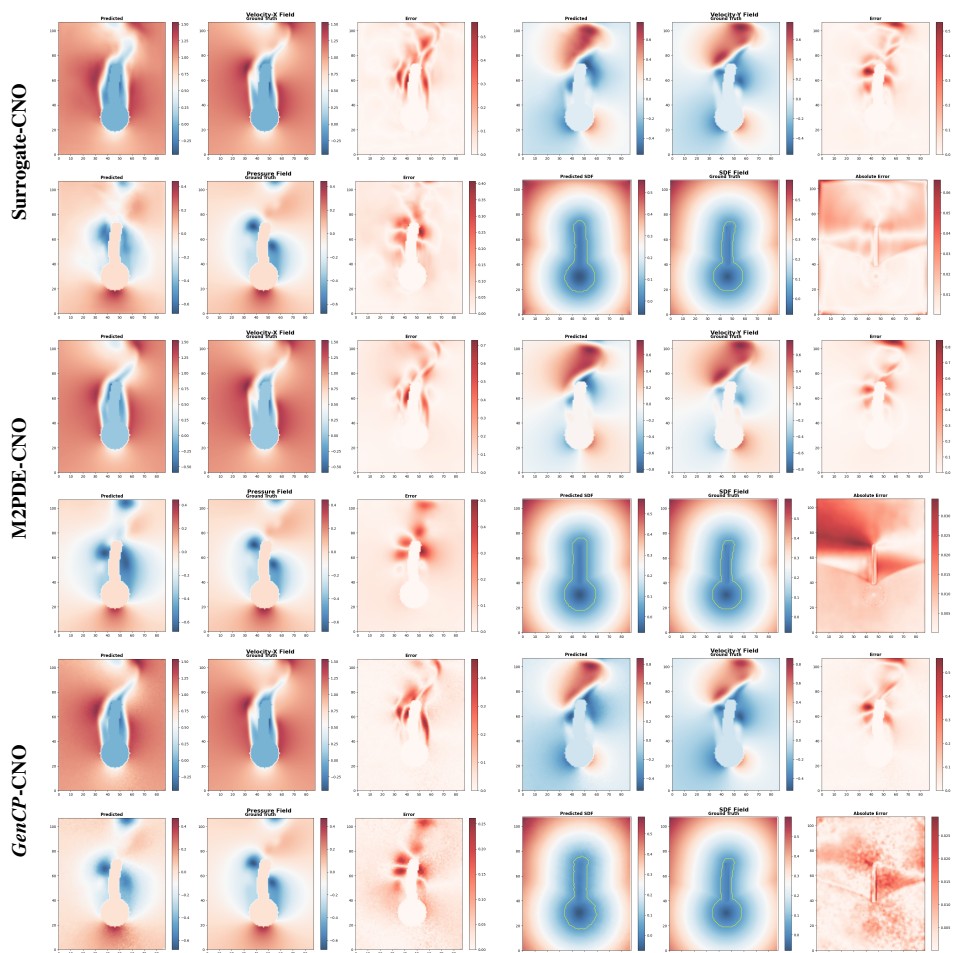

Figure 5: The visualizations of results from different paradigms using CNO as backbone on Turek-Hron setting.

# F    IMPLEMENTATION DETAILS

## F.1    PSEUDOCODE FOR TRAINING

---

**Algorithm 2** Training decoupled conditional velocity fields.

---

1: Input: decoupled datasets $\mathcal{D}_f, \mathcal{D}_g$
2: Initialize $\theta_f, \theta_g$
3: **for** epoch = 1 to $E$ **do**
4:     **for** batch $(f_1, g_0) \sim \mathcal{D}_f$ **do**
5:         Sample $t \sim \mathcal{U}[0,1]$, $z_f \sim \pi_{\mathcal{F}}, z_g \sim \pi_{\mathcal{G}}$
6:         Compute $f_t, g_t, \dot{f}_t$ and update $\theta_f$ to minimize $\|\dot{f}_t - \hat{v}_f(f_t, g_t, t; \theta_f)\|^2$
7:     **end for**
8:     **for** batch $(f_0, g_1) \sim \mathcal{D}_g$ **do**
9:         Sample $t \sim \mathcal{U}[0,1]$, $z'_f \sim \pi_{\mathcal{F}}, z'_g \sim \pi_{\mathcal{G}}$
10:         Compute $f_t, g_t, \dot{g}_t$ and update $\theta_g$ to minimize $\|\dot{g}_t - \hat{v}_g(f_t, g_t, t; \theta_g)\|^2$
11:     **end for**
12: **end for**

---

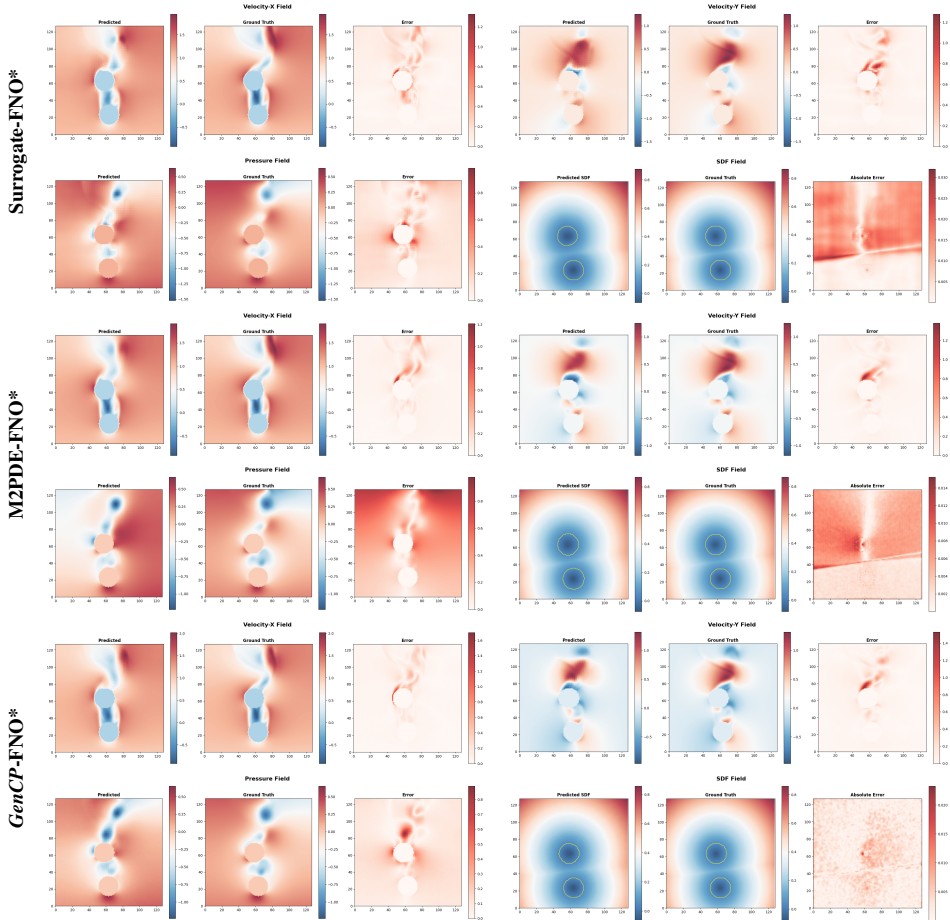

Figure 6: The visualizations of results from different paradigms using FNO* as backbone on Double Cylinder setting.

## F.2 BACKBONE MODELS

This section provides detailed descriptions of the two backbone models used in our *GenCP* framework: Fourier Neural Operator with modifications (FNO*) and Convolutional Neural Operator (CNO). Both models operate in functional spaces, making them particularly suitable for interaction problems where the solution fields are naturally represented as continuous functions over spatial domains.

**FNO*.** The FNO* model operates in the functional space by leveraging the Fourier transform to represent solution fields as combinations of Fourier modes, providing resolution independence, global receptive fields, and alignment with the spectral bias of neural networks toward smooth solutions typical in physical problems. Our implementation combines the Scalable Interpolant Transformer (SiT) framework with Fourier Neural Operator components, where the core computational unit is the spectral convolution layer operating in the Fourier domain as $\mathcal{F}^{-1}(\mathbf{R} \cdot (\mathcal{F}(v)))(\mathbf{x})$, with $\mathbf{R}$ being a learnable linear transformation and the model truncating high-frequency modes by only learning weights for the lowest $M_1 \times M_2 \times M_3$ modes. The architecture incorporates a compact 1D FNO-based final layer that operates along the patch dimension through the sequence $x = \text{OutputProj}(\text{GELU}(\text{BatchNorm}(\text{SpectralConv1d}(x) + \text{Conv1d}(x))))$, along with adaptive layer normalization for timestep conditioning $\text{AdaLN}(x, c) = \gamma(c) \cdot \text{LayerNorm}(x) + \beta(c)$, and both spatial and temporal sinusoidal positional embeddings for rectangular domains and temporal sequences.

Training of the FNO* model employs AdamW optimizer with learning rate $1 \times 10^{-4}$, cosine annealing with warm restarts, batch sizes of 8, gradient clipping with maximum norm

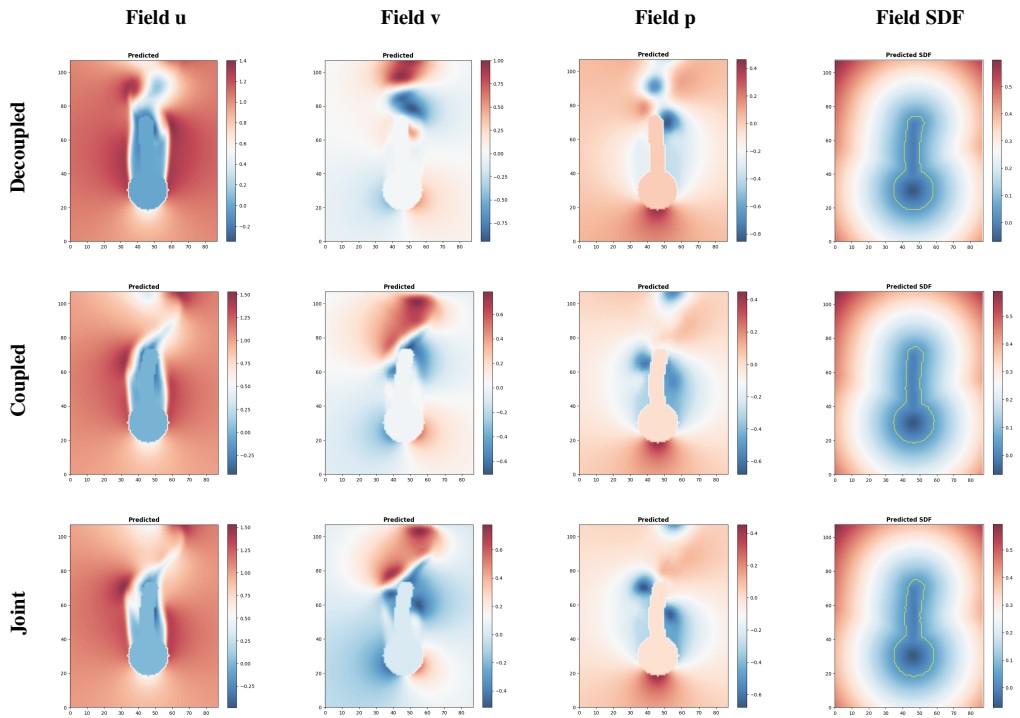

Figure 7: Comparison of results across different training strategies: Decoupled, Coupled, and Joint. Each column corresponds to a physical field (u, v, p, and SDF).

of 1.0, Xavier uniform initialization for linear layers, and zero initialization for output layers to ensure stable convergence. The model processes inputs through spatial patch embedding combined with positional encoding, timestep embedding via sinusoidal encoding followed by MLP transformation, and alternating spatial-temporal transformer blocks where spatial attention operates on $(b \times f, \text{num\_patches}, \text{hidden\_dim})$ tensors while temporal attention processes $(b \times \text{num\_patches}, \text{frames}, \text{hidden\_dim})$ configurations, ultimately producing outputs through the compact FNO final layer that maintains spectral properties while reducing computational complexity through 1D spectral convolutions along the patch dimension.

**CNO.** The CNO model operates in functional space through a hierarchical U-Net-like architecture that preserves the infinite-dimensional nature of solution operators by employing anti-aliased operations, multi-scale processing, and explicit functional lifting and projection operations that map between finite-dimensional representations and infinite-dimensional function spaces. The architecture consists of CNO blocks with filtered Leaky ReLU activations to prevent aliasing during upsampling/downsampling operations, residual blocks incorporating Feature-wise Linear Modulation (FiLM) for timestep conditioning as $\text{FiLM}(x, t) = x \cdot (1 + \gamma(t)) + \beta(t)$, and a lift-project structure where the lifting operation $\text{Lift} : \mathbb{R}^{d_{\text{in}}} \rightarrow \mathbb{R}^{d_{\text{latent}}}$ maps input functions to higher-dimensional latent space while projection $\text{Project} : \mathbb{R}^{d_{\text{latent}}} \rightarrow \mathbb{R}^{d_{\text{out}}}$ maps back to output space. The network processes inputs through five distinct phases: lifting $x_0 = \text{Lift}(u_{\text{input}})$, encoding with downsampling CNO blocks and residual skip connections $x_{i+1} = \text{CNOBlock}^{(D)}(x_i)$ and $s_i = \text{ResBlock}^{N_{\text{res}}}(x_i)$, bottleneck processing $x_{\text{neck}} = \text{ResBlock}^{N_{\text{neck}}}(x_{\text{deepest}})$, decoding with upsampling and optional invariant blocks $x_i = \text{CNOBlock}^{(U)}(\text{Concat}(x_i, s_{N-i}))$, and final projection $u_{\text{output}} = \text{Project}(\text{Concat}(x_0, s_0))$.

Training of the CNO model utilizes Adam optimizer with learning rate $2 \times 10^{-4}$, exponential decay scheduler with $\gamma = 0.99$, batch size of 8 for 3D volumes, batch normalization without dropout for regularization, and MSE loss with gradient penalty for stability. The model integrates timestep information through sinusoidal encoding $t_{\text{emb}} = \text{MLP}([\sin(\omega_0 t), \cos(\omega_0 t), \ldots, \sin(\omega_{d/2} t), \cos(\omega_{d/2} t)])$ and employs carefully designed anti-aliasing filters with cutoff frequency $\omega_c = s/2.0001$ slightly

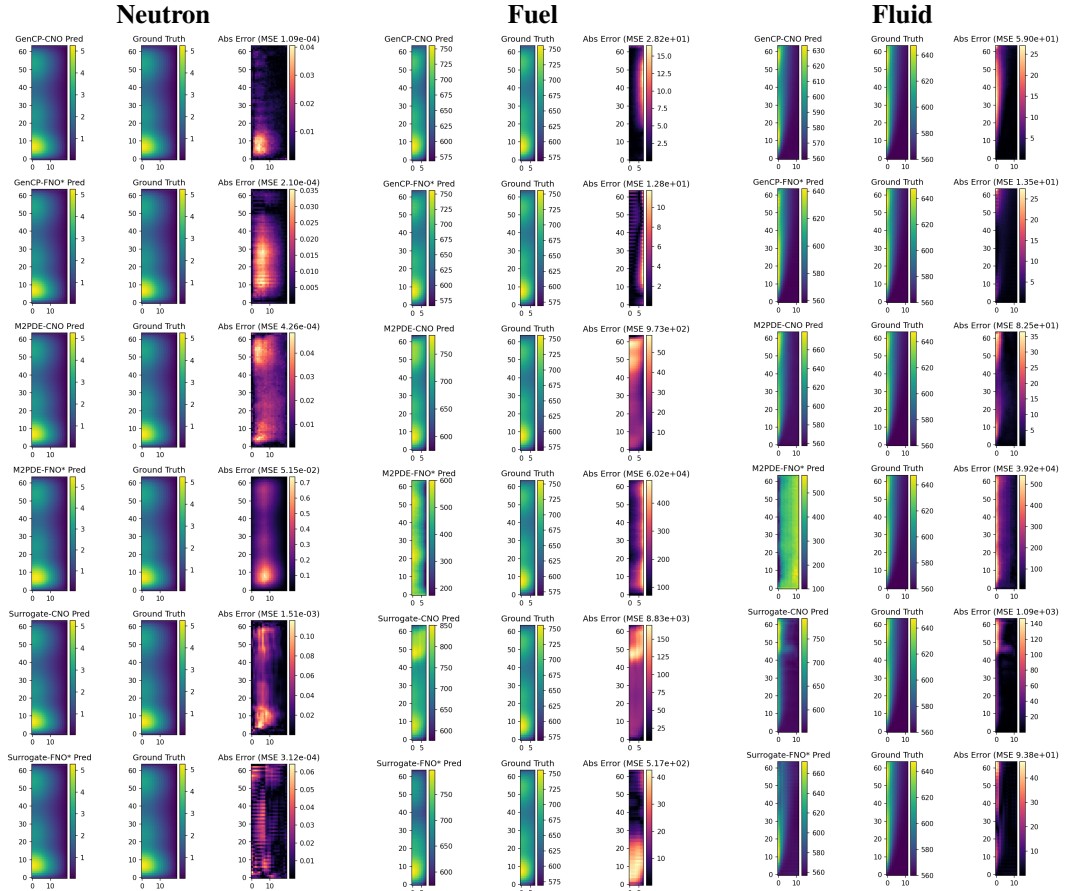

Figure 8: The visualization of the final frame of predicted results from different paradigms on NT Coupling setting. (Here we only represent $T_f$ as fluid field visualization.)

below the Nyquist frequency, half-width $\Delta = 0.8s - s/2.0001$, 6-tap Kaiser window filter design, and upsampling factor of 2 for LReLU operations to maintain the continuous function interpretation throughout the hierarchical processing while capturing both fine-grained local features and global structure necessary for accurate multiphysics interaction modeling.

### F.3 HYPERPARAMETERS OF BACKBONE MODELS

To ensure a fair comparison and minimize bias from hyperparameter tuning, we standardized the backbone hyperparameters for each task. Minor adjustments were made only to accommodate the inherent modeling complexity of different methods, as detailed in Tabel 5, 6, 7, 8, 9, 10.

Table 5: Hyperparameters of *GenCP*-FNO* for different datasets.

|  | Turek Hron | | Double Cylinder | | NT Coupling | | |
| --- | --- | --- | --- | --- | --- | --- | --- |
| Hyperparameter name | fluid | structure | fluid | structure | fluid | fuel | neutron |
| Number of Blocks | 4 | 4 | 4 | 4 | 4 | 4 | 4 |
| FNO mode | 4 | 4 | 4 | 4 | 4 | 4 | 4 |
| Hidden size | 256 | 128 | 256 | 128 | 256 | 128 | 128 |
| Patch size | [2, 2] | [2, 2] | [2, 2] | [2, 2] | [2, 2] | [2, 2] | [2, 2] |
| Number of blocks | 4 | 4 | 4 | 4 | 4 | 4 | 4 |
| MLP ratios | 4 | 4 | 4 | 4 | 4 | 4 | 4 |

Table 6: Hyperparameters of M2PDE-FNO* for different datasets.

| Hyperparameter name | Turek Hron | | Double Cylinder | | NT Coupling | | |
| --- | --- | --- | --- | --- | --- | --- | --- |
| | fluid | structure | fluid | structure | fluid | fuel | neutron |
| Number of Blocks | 4 | 4 | 4 | 4 | 4 | 4 | 4 |
| FNO mode | 4 | 4 | 4 | 4 | 4 | 4 | 4 |
| Hidden size | 256 | 128 | 256 | 128 | 256 | 128 | 128 |
| Patch size | [2, 2] | [2, 2] | [2, 2] | [2, 2] | [2, 2] | [2, 2] | [2, 2] |
| Number of blocks | 4 | 4 | 4 | 4 | 4 | 4 | 4 |
| MLP ratios | 4 | 4 | 4 | 4 | 4 | 4 | 4 |

Table 7: Hyperparameters of Surrogate-FNO* for different datasets.

| Hyperparameter name | Turek Hron | | Double Cylinder | | NT Coupling | | |
| --- | --- | --- | --- | --- | --- | --- | --- |
| | fluid | structure | fluid | structure | fluid | fuel | neutron |
| Number of Blocks | 4 | 4 | 4 | 4 | 4 | 4 | 4 |
| FNO mode | 4 | 4 | 4 | 4 | 4 | 4 | 4 |
| Hidden size | 128 | 64 | 128 | 64 | 128 | 64 | 64 |
| Patch size | [2, 2] | [2, 2] | [2, 2] | [2, 2] | [2, 2] | [2, 2] | [2, 2] |
| Number of blocks | 4 | 4 | 4 | 4 | 4 | 4 | 4 |
| MLP ratios | 4 | 4 | 4 | 4 | 4 | 4 | 4 |

Table 8: Hyperparameters of *GenCP*-CNO for different datasets.

| Hyperparameter name | Turek Hron | | Double Cylinder | | NT Coupling | | |
| --- | --- | --- | --- | --- | --- | --- | --- |
| | fluid | structure | fluid | structure | fluid | fuel | neutron |
| Number of layers | 4 | 2 | 4 | 2 | 4 | 2 | 2 |
| Number of residual blocks per level | 1 | 1 | 1 | 1 | 1 | 1 | 1 |
| Number of residual blocks in the neck | 6 | 6 | 6 | 6 | 6 | 6 | 6 |
| Channel Multiplier | 16 | 16 | 16 | 16 | 16 | 16 | 16 |
| Size of Conv Kernel | 3 | 3 | 3 | 3 | 3 | 3 | 3 |
| latent dimension in the lifting layer | 64 | 64 | 64 | 64 | 64 | 64 | 64 |

Table 9: Hyperparameters of M2PDE-CNO for different datasets.

| Hyperparameter name | Turek Hron | | Double Cylinder | | NT Coupling | | |
| --- | --- | --- | --- | --- | --- | --- | --- |
| | fluid | structure | fluid | structure | fluid | fuel | neutron |
| Number of layers | 4 | 2 | 4 | 2 | 4 | 2 | 2 |
| Number of residual blocks per level | 1 | 1 | 1 | 1 | 1 | 1 | 1 |
| Number of residual blocks in the neck | 6 | 6 | 6 | 6 | 6 | 6 | 6 |
| Channel Multiplier | 16 | 16 | 16 | 16 | 16 | 16 | 16 |
| Size of Conv Kernel | 3 | 3 | 3 | 3 | 3 | 3 | 3 |
| latent dimension in the lifting layer | 64 | 64 | 64 | 64 | 64 | 64 | 64 |

Table 10: Hyperparameters of Surrogate-CNO for different datasets.

| Hyperparameter name | Turek Hron | | Double Cylinder | | NT Coupling | | |
|---|---|---|---|---|---|---|---|
| | fluid | structure | fluid | structure | fluid | fuel | neutron |
| Number of layers | 4 | 2 | 4 | 2 | 4 | 2 | 2 |
| Number of residual blocks per level | 1 | 1 | 1 | 1 | 1 | 1 | 1 |
| Number of residual blocks in the neck | 6 | 6 | 6 | 6 | 6 | 6 | 6 |
| Channel Multiplier | 8 | 8 | 8 | 8 | 8 | 8 | 8 |
| Size of Conv Kernel | 3 | 3 | 3 | 3 | 3 | 3 | 3 |
| latent dimension in the lifting layer | 64 | 64 | 64 | 64 | 64 | 64 | 64 |

## F.4 INFERENCE SCHEME OF BASELINE METHODS

**Surrogate-based scheme**   The surrogate model in this article refers to a neural network that directly maps system inputs to outputs in one step, unlike diffusion models that generate outputs step-by-step through denoising. For multiphysics problems, surrogate models are trained separately for each physical field using decoupled data and obtain coupled solutions through iteration.

Our work follows the convention from M2PDE (Zhang et al., 2025) for implementing the surrogate model combination algorithm in a Picard-iteration approach. The surrogate models' combination algorithm are identical, as demonstrated in Alg 3. The relaxation factor $\alpha$ is set to 0.5 by default, and the tolerance $\epsilon_{\max}$ is set to 0 by default, effectively running the iteration for the maximum number of steps without early stopping based on convergence.

---

**Algorithm 3** Surrogate Model Combination Algorithm for Multiphysics Simulation.

---

**Require:** Compositional set of surrogate model $\epsilon_\theta^i(z_{\neq i}, C)$, $i = 1, 2, \ldots, N$, outer inputs $C$, maximum number of iterations $M$, tolerance $\epsilon_{\max}$, relaxation factor $\alpha$.
1:   Initialize constant fields $z_i$, $i = 1, \ldots, N$, $m = 0$
2:   **while** $m < M$ and $\epsilon > \epsilon_{\max}$ **do**
3:       $m = m + 1$
4:       **for** $i = 1, \ldots, N$ **do**
5:           $\hat{z}_i = z_i$
6:           $z_i = \alpha \epsilon_\theta^i(z_{\neq i}, C) + (1 - \alpha)\hat{z}_i$
7:       **end for**
8:       $\epsilon = L1(z_i - \hat{z}_i)$, $i = 1, \ldots, N$
9:   **end while**
**Ensure:** $z_1, z_2, \ldots, z_N$

---

The notation used in Algorithm 3 differs from this paper's notations and is briefly explained as follows: $\epsilon_\theta^i$ represents the trained surrogate model, such as a neural network like FNO, for the $i$-th physical field; $z_{\neq i}$ denotes the predictions from all other physical fields used as conditions; $C$ represents the outer system inputs, such as initial conditions or boundary values; $z_i$ are the predicted fields for each physical process; $\hat{z}_i$ is a temporary copy of the previous prediction; $\alpha$ is the relaxation factor, a scalar between 0 and 1; $M$ is the maximum number of iterations; and $\epsilon_{\max}$ is the convergence tolerance threshold. The L1 norm ($L1$) computes the absolute difference as a convergence metric, which can be adapted to other norms like L2 if needed.

The algorithm takes trained single-field surrogate models, one per physical process and trained on decoupled data, and iteratively combines them to approximate a coupled multiphysics solution. It starts with initial constant fields $z_i$, such as a uniform value like 0.5 or problem-specific priors, along with outer inputs $C$. In each iteration, it updates each field's prediction conditioned on the others and applies a moving average for stabilization. The loop ends when the maximum iterations $M$ are reached or the convergence metric $\epsilon$, based on the L1 norm of prediction changes, falls below $\epsilon_{\max}$.

The moving average with relaxation factor $\alpha$ prevents divergence or instability in iterative updates, particularly in strongly coupled systems where direct updates ($\alpha = 1$) might oscillate or fail to converge. This approach resembles under-relaxation in traditional numerical methods like Picard

iteration, blending new predictions with previous ones weighted by $\alpha$ and $1 - \alpha$ to ensure smoother convergence.

**M2PDE scheme** The M2PDE framework composes multiple diffusion models, each trained on a single physical process using decoupled data, to simulate coupled multiphysics systems through iterative conditional sampling during inference. Unlike surrogate models that directly map inputs to outputs, M2PDE generates solutions via a reverse diffusion process, implicitly handling interactions between fields.

---

**Algorithm 4** Inference of Multiphysics Simulation with the Composition of Diffusion Models.

---

**Require:** Compositional set of diffusion models $\epsilon_\theta^i(z_{\neq i}, C, t)$, $i = 1, 2, \ldots, N$, outer inputs $C$, number of reverse diffusion steps $T$, number of outer iterations $K$.

1: **for** $k = 1, \ldots, K$ **do**
2: $\quad z_T \sim \mathcal{N}(0, I)$
3: $\quad$ **for** $t = T, \ldots, 1$ **do**
4: $\quad\quad$ **for** $i = 1, \ldots, N$ **do**
5: $\quad\quad\quad \hat{z}_i^0 = \epsilon_\theta^i(z_{\neq i}^t, C, t)$ $\qquad\qquad\qquad$ ▷ Estimate clean sample for component $i$
6: $\quad\quad$ **end for**
7: $\quad\quad$ Compose $\hat{z}^0 = (\hat{z}_1^0, \ldots, \hat{z}_N^0)$
8: $\quad\quad z^{t-1} = p_\theta(z^t | \hat{z}^0, t)$ $\qquad\qquad\qquad$ ▷ Perform conditional sampling step
9: $\quad$ **end for**
10: $\quad z^{(k)} = z^0$
11: **end for**
**Ensure:** $z_1^{(K)}, z_2^{(K)}, \ldots, z_N^{(K)}$

---

The composition algorithm is identical, as demonstrated in Alg. 4. The number of outer iterations $K$ is set to 2 by default. As introduced in the surrogate model combination algorithm, M2PDE schema shares the same notations: $\epsilon_\theta^i$ represents the trained diffusion model (score network) for the $i$-th physical field; $z_{\neq i}^t$ denotes the current noisy estimates from all other physical fields at diffusion timestep $t$; $C$ represents the outer system inputs, such as initial conditions or boundary values; $z^t$ is the noisy state at timestep $t$; $\hat{z}_i^0$ is the estimated clean (denoised) sample for field $i$; $T$ is the total number of reverse diffusion steps; $K$ is the number of outer iterations; and $p_\theta$ denotes the conditional sampling function in the diffusion process.

The algorithm takes trained single-field diffusion models, one per physical process and trained on decoupled data, and iteratively composes them during reverse diffusion to generate a coupled multiphysics solution. It starts by sampling initial noise $z_T$ from a standard normal distribution, along with outer inputs $C$. In each outer iteration $k$, it performs a full reverse diffusion process: for each timestep $t$ from $T$ to 1, it estimates denoised samples for each field conditioned on the others' current states, composes them, and advances the sampling step. The outer loop refines the solution over $K$ iterations, with the final output being the denoised fields after the last iteration.

The outer iterations ($K$) enable progressive refinement of field interactions, improving accuracy in strongly coupled systems. Conditional sampling ensures that each field's generation accounts for others, implicitly capturing multiphysics couplings without explicit iteration during training.

## G   LIMITATIONS AND FUTURE WORK

*GenCP* has been evaluated on two-dimensional synthetic distribution and multiphysical problems, without yet being applied to more complex systems like 3D or real-world engineering problems. Additionally, it has not considered irregular geometric boundaries, which is another notable limitation. These limitations arise because we primarily focus on introducing our core paradigm, and are further constrained by limited data availability, which also opens up exciting and broad avenues for future research. Besides, we have not yet explored incorporating physical priors into our framework. Such integration may further improve our performance on out-of-distribution data, thereby enhancing both the accuracy and robustness of coupled predictions after training on decoupled data. Looking ahead, we also plan to leverage *GenCP* to unlock the value of abundant historical single-physics data and integrate cross-institutional datasets for learning complex coupled physics.

