# OpenReview forum: "GenCP: Towards Generative Modeling Paradigm of Coupled physics"
_ICLR.cc/2026/Conference — ICLR 2026 Poster_

### Official Review · Reviewer_CPwt · 2025-10-26

**Soundness:** 3
**Presentation:** 3
**Contribution:** 4
**Rating:** 8
**Confidence:** 3

**Summary:**

This paper presents GenCP, a novel paradigm for multiphysics simulation that learns coupled physics from decoupled training data.

**Key Innovation:** The elegant combination of flow matching's ODE formulation with operator splitting theory. The method trains separate flow matching models on decoupled data (learning $p(f|g)$ and $p(g|f)$), then composes them via Lie-Trotter splitting during inference to generate coupled solutions $p(f,g)$.

**Theoretical Contribution:** Establishes rigorous convergence guarantees by connecting probability density evolution with operator splitting.

**Significance:** This work bridges generative models with classical numerical methods, providing both theoretical rigor and practical efficiency for solving expensive coupled simulation problems.

**Strengths:**

- **Intuitive decomposition with strong empirical validation.** The core idea of learning coupled physics through separate conditional models is elegant and practical. The paper convincingly demonstrates this works: models trained only on decoupled data successfully capture strongly coupled dynamics, which surrogate baselines completely fail to model despite similar numerical errors. This validates that the probabilistic formulation genuinely captures coupling mechanisms.

- **Rigorous theoretical foundation bridging generative models and numerical analysis.** The connection between flow matching's velocity field decomposition ($v = v^{(f)} + v^{(g)}$) and Lie-Trotter operator splitting is non-trivial and principled. The formal convergence analysis via Wasserstein-1 distance provides error controllability guarantees rare in neural PDE solvers, with explicit tracking of splitting error ($\tau$) and learning error ($\varepsilon_f, \varepsilon_g$).

- **Careful decoupled data construction avoiding information leakage.** The experimental design modifies boundary conditions to ensure zero coupling information in training data while maintaining physical plausibility. This proves the method learns underlying conditional physics rather than memorizing coupling patterns.

- **Insightful analysis of baseline failure modes.** Beyond reporting metrics, the paper explains why deterministic surrogate methods fail: they miss oscillatory dynamics despite low numerical errors because they cannot handle mode errors and stochastic behavior. This clarifies that GenCP's probabilistic modeling addresses fundamental limitations of deterministic learning for strongly coupled systems.

**Weaknesses:**

- **Notation density limits accessibility.** The formulation rapidly introduces weak continuity equations and Lie-Trotter splitting without sufficient intuitive scaffolding. Key notations need clearer explanation before formal treatment.

- **Incomplete Decoupling.** The experiments acknowledge (Appendix D) that one training direction uses coupled rather than truly decoupled data due to negligible structural deformation in genuinely decoupled scenarios. This "half-decoupled" compromise contradicts the paper's emphasis on training exclusively from decoupled data, suggesting the method still requires partial access to expensive coupled simulations in practice.

**Questions:**

- The paper assumes decoupled data is easier to obtain (line 47), but your experimental protocol suggests otherwise. You still need to run solvers to generate it. Would it be more accurate to position GenCP's value as enabling the "reuse" of historical single-physics data or "integration" of cross-institutional datasets, rather than claiming acquisition cost advantages in general?

- Experimental Design: The paper acknowledges in Appendix D that structure-conditioned-on-fluid training uses coupled rather than truly decoupled data. Could the authors provide ablation studies comparing: (1) fully decoupled training for both directions, (2) the current "half-decoupled" setting, and (3) fully coupled training? This would help quantify the performance trade-offs and clarify whether truly decoupled training is feasible for both directions, which is central to the paper's claims.

- The success of GenCP relies on the assumption that decoupled datasets $\mathcal{D}_f$ and $\mathcal{D}_g$ contain sufficient information to recover the joint distribution. In your experiments, this is guaranteed by construction, since both datasets are derived from the same coupled solver with consistent physical parameters. Can you characterize the necessary and sufficient conditions on decoupled data for GenCP to work? Specifically:
  - Must $\mathcal{D}_f$ and $\mathcal{D}_g$ share the same physical parameters?
  - What level of parameter mismatch is tolerable between the two datasets?
  - Are there failure modes where learning $p(f|g)$ and $p(g|f)$ separately is provably insufficient to reconstruct the joint distribution $p(f,g)$?
  - How can practitioners assess the "coupling information completeness" of their real-world decoupled datasets before committing to training?

---

> ### Author Response · Authors · 2025-11-22
> **Official Response to Reviewer CPwt (1)**
>
> We greatly appreciate your recognition of our article, particularly regarding the "rigorous theoretical foundation bridging generative models and numerical analysis", exactly what we aimed to achieve. Meanwhile, in response to the reviewer comments, we have **supplemented the intuitive explanations** in the revised manuscript, and strictly **verified our algorithm on fully decoupled data** through the supplemented Nuclear-Thermal Coupling experiments.
>
> > W1: Notation density limits accessibility. The formulation rapidly introduces weak continuity equations and Lie-Trotter splitting without sufficient intuitive scaffolding. Key notations need clearer explanation before formal treatment.
> >
>
> A: Thank you for your reminder. We have supplemented the main text with more intuitive explanations and more specific symbolic illustrations in the revised manuscript. The specific content is as follows:
>
> "We first recall the strong-form continuity equation. If the joint probability law $\mu_t$ admits a density $\rho_t(u)$ on the functional state space $\mathcal{U} = \mathcal{F} \times \mathcal{G}$, and if $\rho_t$ is sufficiently regular, then its evolution under a time-dependent velocity field $v(t,u)$ satisfies the strong-form PDE
>
> $$\partial_t \rho_t(u) + \nabla_u \cdot \big( \rho_t(u)\, v(t,u) \big) = 0,$$
>
> expressing conservation of probability mass as it is transported by $v$. In finite dimensions this PDE is well-defined, but for distributions over function spaces several obstacles arise: empirical measures coming from decoupled physics solvers are typically singular and do not admit a density $\rho_t$; the divergence operator $\nabla_u\cdot(\rho_t v)$ is not meaningful when $u$ represents functions rather than vectors; and even when densities exist, numerical or learned models cannot reliably approximate derivatives in infinite-dimensional spaces."
>
> "The Lie--Trotter splitting scheme provides a principled mechanism for recombining the two learned partial dynamics during inference.  Because $v = v^{(f)} + v^{(g)}$, the full coupled evolution can be approximated over a small time step $\tau$ by first evolving $f$ with $g$ held fixed (the flow induced by $v^{(f)}$), and then evolving $f$ with $g$ held fixed (the flow induced by $v^{(g)}$).  In the limit of small $\tau$, this alternating update is consistent with the joint flow generated by $v^{(f)} + v^{(g)}$, which explains how separately learned conditional velocities can be merged into a coherent coupled inference procedure."

---

> ### Author Response · Authors · 2025-11-22
> **Official Response to Reviewer CPwt (2)**
>
> > W2: Incomplete Decoupling. The experiments acknowledge (Appendix D) that one training direction uses coupled rather than truly decoupled data due to negligible structural deformation in genuinely decoupled scenarios. This "half-decoupled" compromise contradicts the paper's emphasis on training exclusively from decoupled data, suggesting the method still requires partial access to expensive coupled simulations in practice.
> >
>
> A: Thanks for the Reviewer's comment.
>
> At first, we wish to clarify that the 'half-decoupled' experimental design does not undermine our core algorithmic contribution regarding conditional-to-joint probabilistic sampling. As formalized by the error bound in Theorem 3.1, the total error of our generated joint distribution comprises two distinct components: the splitting error $\tau$, arising from probability density evolution, and the learning error $\epsilon$, representing the approximation gap between the learned and true conditional vector fields. The use of half-decoupled data in the original experiments merely simplified the learning task for $\epsilon$ (specifically for the structural field), but it did not alter or simplify the splitting mechanism $\tau$, which constitutes the core innovation of our algorithm. Therefore, the **theoretical validity of our 'conditional-to-joint' sampling paradigm remains intact**, regardless of the specific data complexity used for demonstration.
>
> To further address the Reviewer's concern that the use of "half-decoupled" data might compromise the validity of our approach, we have conducted an **additional experiment** using the Nuclear-Thermal (NT) Coupling setting, which **relies strictly on fully decoupled data**. Since the detailed setup is provided in Appendix F, we briefly summarize it here: this scenario involves three distinct physical fields (neutron, fluid, and solid) characterized by highly complex interactions, including regional and interface coupling, strong and weak coupling, as well as unidirectional and bidirectional coupling. Our objective was to train models exclusively on fully decoupled data from these three isolated single fields and subsequently infer the coupled physical trajectory of the entire system over time, given only external boundary conditions.
>
> We benchmarked the GenCP, M2PDE, and surrogate-based paradigms on this task. Detailed quantitative analyses and visualizations are available in Appendix F, and the validation errors on decoupled data and testing errors on coupled data are summarized in the table below. As shown, GenCP significantly outperforms the other two baselines across all three fields, achieving error reductions exceeding 50%. These results compellingly demonstrate that our method **maintains superior performance** even when using strictly decoupled training data.
>
> | Rel L2 Norm | Validation on decoupled data | Validation on decoupled data | Validation on decoupled data | Test on coupled data | Test on coupled data | Test on coupled data |
> | --- | --- | --- | --- | --- | --- | --- |
> | Field | Neutron | Solid | Fluid | Neurton | Solid | Fluid |
> | GenCP-FNO* | 0.0022 | 0.0006 | 0.0038 | **0.0085** | **0.0364** | **0.0270** |
> | Surrogate-FNO* | 0.0086 | 0.0014 | 0.0032 | 0.0149 | 0.0576 | 0.1095 |
> | M2PDE-FNO* | 0.0052 | 0.0014 | 0.0018 | 0.0136 | 0.1237 | 0.0463 |
> | GenCP-CNO | 0.0024 | 0.0005 | 0.0110 | **0.0044** | **0.0105** | **0.0330** |
> | Surrogate-CNO | 0.0046 | 0.0007 | 0.0082 | 0.0130 | 0.0553 | 0.3086 |
> | M2PDE-CNO | 0.0053 | 0.0016 | 0.0092 | 0.0164 | 0.0646 | 0.0401 |

---

> ### Author Response · Authors · 2025-11-22
> **Official Response to Reviewer CPwt (3)**
>
> > Q1: The paper assumes decoupled data is easier to obtain (line 47), but your experimental protocol suggests otherwise. You still need to run solvers to generate it. Would it be more accurate to position GenCP's value as enabling the "reuse" of historical single-physics data or "integration" of cross-institutional datasets, rather than claiming acquisition cost advantages in general?
> >
>
> A: We appreciate the reviewer's excellent suggestion, which helps to significantly broaden the impact of our work. We strongly agree that enabling the "reuse" of historical single-physics data and the "integration" of cross-institutional datasets to model complex coupled physics are **compelling value propositions** for GenCP. Accordingly, we have incorporated these insights into the Future Work Section in the revised manuscript to emphasize them as potential benefits of our paradigm.
>
> However, we respectfully clarify that the necessity of running solvers does not contradict the claim of lower data acquisition costs. Although numerical solvers are employed in both scenarios, the **computational complexity and stability challenges associated with generating decoupled data are significantly lower** than those for coupled data. Specifically, generating coupled data via monolithic methods requires substantially huge computational overhead [1], while partitioned methods typically require 5 to 20 iterative convergence checks (sub-iterations) at every time step to ensure stability [2, 3], whereas decoupled generation runs in only one single stable pass. Therefore, we maintain that "decoupled data is easier to obtain" represents a valid advantage.
>
> Reference:
>
> [1] Hron, Jaroslav, and Stefan Turek. "A monolithic FEM/multigrid solver for an ALE formulation of fluid-structure interaction with applications in biomechanics." *Fluid-Structure Interaction: Modelling, Simulation, Optimisation*. Berlin, Heidelberg: Springer Berlin Heidelberg, 2006. 146-170.
>
> [2] Degroote, Joris, et al. "Stability of a coupling technique for partitioned solvers in FSI applications." *Computers & Structures* 86.23-24 (2008): 2224-2234.
>
> [3] Causin, Paola, Jean-Frédéric Gerbeau, and Fabio Nobile. "Added-mass effect in the design of partitioned algorithms for fluid–structure problems." *Computer methods in applied mechanics and engineering* 194.42-44 (2005): 4506-4527.
>
> > Q2: Experimental Design: The paper acknowledges in Appendix D that structure-conditioned-on-fluid training uses coupled rather than truly decoupled data. Could the authors provide ablation studies comparing: (1) fully decoupled training for both directions, (2) the current "half-decoupled" setting, and (3) fully coupled training? This would help quantify the performance trade-offs and clarify whether truly decoupled training is feasible for both directions, which is central to the paper's claims.
> >
>
> A: We thank the Reviewer for the constructive suggestions regarding additional ablation studies. We have addressed each point as follows:
>
> 1. Regarding Suggestion (2): We clarify that the experimental setting described just corresponds exactly to the primary methodology already implemented in our paper.
> 2. Regarding Suggestion (3): We had **previously conducted** this analysis (joint training on coupled data) for the Double Cylinder setting, as shown in Table 3 in the manuscript. Following your recommendation, we have now **extended** this to the Turek-Hron dataset and added the fully coupled training results to Table 2 in the manuscript. As expected, training on coupled data (if available) theoretically yields the upper bound of performance. However, since coupled data is often computationally prohibitive or unavailable, the value of GenCP lies in its ability to infer coupled solutions from accessible decoupled data.
> 3. Regarding Suggestion (1): This was indeed our original intention. However, as detailed in Appendix D and our response to W2, inherent data limitations (specifically the lack of structural motion in the fluid-conditioned subset) prevented a meaningful evaluation in that specific setting. Nevertheless, to robustly demonstrate our core claim that GenCP enables coupled inference from purely decoupled data, we have conducted an additional experiment on the NT Coupling Dataset (details have been introduced in W2).

---

> ### Author Response · Authors · 2025-11-22
> **Official Response to Reviewer CPwt (4)**
>
> > Q3: The success of GenCP relies on the assumption that decoupled datasets  and  contain sufficient information to recover the joint distribution. In your experiments, this is guaranteed by construction, since both datasets are derived from the same coupled solver with consistent physical parameters. Can you characterize the necessary and sufficient conditions on decoupled data for GenCP to work? Specifically:
> >
> > - Must $D_f$ and $D_g$ share the same physical parameters?
> > - What level of parameter mismatch is tolerable between the two datasets?
> > - Are there failure modes where learning $p(f|g)$ and $p(g|f)$ separately is provably insufficient to reconstruct the joint distribution ?
> > - How can practitioners assess the "coupling information completeness" of their real-world decoupled datasets before committing to training?
>
> A: We thank the Reviewer for this very insightful question, which touches upon the core necessary conditions for our paradigm to function.
>
> Before addressing the specific sub-questions, we respectfully clarify the experimental setup. We did not use the "same coupled solver" for both training and testing. The coupled test data is generated by a standard bidirectional FSI solver where fluid and structure interact iteratively. In contrast, the decoupled training data is generated by unidirectional solvers. Therefore, the success of **GenCP relies not on using the "same solver," but on the insight that if the decoupled datasets sufficiently cover the underlying physical principles** (specifically the conditional probabilities $p(f|g)$ and $p(g|f)$), then the joint physics $p(f,g)$ can be recovered.
>
> Below, we address your four specific questions:
>
> 1. Yes, in the current implementation. Since our current model is **not explicitly conditioned on physical parameters** (e.g., Reynolds number or Young's modulus) as inputs, the training datasets must share consistent physical parameters with the test data to ensure the learned conditional distributions are compatible. However, if the model were trained to generalize across parameters (by taking physical parameters as input tokens), exact parameter matching would not be required during inference.
> 2. It is difficult to quantify a universal tolerance threshold for parameter mismatch, as this depends on the sensitivity of the specific physical system. Intuitively, the key issue is not "mismatch", but Out-of-Distribution (OOD) behavior. Small mismatches typically introduce a minor shift in the conditional manifolds, which could be tolerated. While, significant mismatches would push the inference process into an OOD regime where the learned conditional priors are no longer valid, leading to performance degradation or even failure.
> 3. Honestly, yes. A fundamental failure mode is incomplete supporting data, where the physics contained in the decoupled training data does not cover the states visited by the true coupled solution. As discussed in Appendix D, if we were to use a strictly "solid-condition-on-fluid" dataset where the beam never bends, the model would learn a conditional distribution $p(g|f)$ that is nearly a Dirac delta at zero deformation, which could also be interpreted as OOD. Even if the fluid model $p(f|g)$ is perfect, the joint sampler could never recover the large-deformation coupled solution because the structural model has never learned the physics of bending. This again confirms that the decoupled priors must have adequate support over the relevant physical behaviors.
> 4. To assess completeness, practitioners should conduct a physically grounded case-by-case analysis rather than relying on a single statistical metric. The core criterion is whether the input conditions and output responses in the **decoupled datasets sufficiently encompass the physical essences expected in the coupled scenario**. As long as the decoupled datasets sufficiently encode these essential physical laws (the conditional responses) within the region of interest, GenCP can mathematically recombine them into a valid joint trajectory.

---

> > ### Comment · Reviewer_CPwt · 2025-11-24
> >
> > My concerns have been fully resolved, and I have no further questions. Based on the authors' clarifications, I will maintain my initial score.

---

> > > ### Author Response · Authors · 2025-11-24
> > > **Official Response to Reviewer CPwt**
> > >
> > > Thank you for your constructive feedback throughout the review process. We are pleased that the additional experiments and theoretical clarifications have met your expectations. Your suggestions have significantly strengthened our paper, and we sincerely appreciate your support and the positive rating.

---

### Official Review · Reviewer_Uo4f · 2025-10-31

**Soundness:** 2
**Presentation:** 2
**Contribution:** 2
**Rating:** 2
**Confidence:** 4

**Summary:**

The authors introduces GenCP, a generative modeling framework for simulating coupled multiphysics systems with applications to fluid-structure interaction. Their core contribution is training on decoupled data which does not require coupled simulation data during the training process. During inference,however, the  fully coupled system can be predicted.

**Strengths:**

- According to the Reviewer, the general idea presented in the paper of integrating operator-splitting theory with generative flow models is novel. Using decoupled training and coupled inference later to reduce the cost of training data could potentially be very impactful as many AI4Science ML applications deal with the issue of spending significant compute on training data generation.

- The authors present theoretical guarantees for their framework. The method is not just a heuristic but directly connected to operator splitting theory used in numerical schemes.

**Weaknesses:**

Despite the core idea being novel, the Reviewer is unable to recommend the paper for acceptance at ICLR due to the following major issues:
- One of the central motivation of the paper is that the training data generation using decoupled systems only is cheaper but this was unfortunately never quantified. The reviewer recommends evaluating the cost of training data generation for coupled and decoupled generation.
- The comparisons to baselines should also include frameworks trained on the fully coupled data to estimate the change in accuracy by using decoupled data only.
- The authors only apply their method to, in the setting of FSI, small problems. However, as stated in the Appendix for one of their examples the authors do not use decoupled data only but coupled training data as the decoupled data contained no meaningful structural motion. According to the Reviewer, this undermines one of the central statement of the paper. If already for a small example, using decoupled training data only is insufficient, the Reviewer would expect for more complex system with more non-linear coupling the same issue. This would significantly limit the applicability of the new approach.
- The obtained relative errors on the coupled data test set seem relatively high and would be too high for most engineering applications. The Reviewer acknowledges that the authors perform better than reported baselines but the gap between results on the decoupled validation set and the coupled test set seems to high and would discourage currently training on decoupled data in the Reviewers opinion. Moreover, the Reviewer encourages the authors to add the aforementioned baselines on a fully coupled dataset.
- The authors use convolutional building blocks and do therefore require a regular grid. For FSI which often involves complex boundaries this is a significant limitation. In Figure 6, the cylinder is not round but the discretization artifacts are clearly visible. The authors should comment on this limitation.

**Questions:**

See weaknesses and:
- For the two FSI examples, the authors just report single error values but the Reviewer would have expected a distribution due to the probabilistic approach. Are these posterior mean values ?

---

> ### Author Response · Authors · 2025-11-22
> **Official Response to Reviewer Uo4f (1)**
>
> We would like to thank you for your constructive suggestions and questions regarding our method. In response, we have made systematic supplements and clarifications in the revised manuscript. We clarify that **"semi-decoupled" data does not affect the core** splitting mechanism, and **verify the effectiveness of GenCP in complex systems** with supplmented Nuclear-Thermal Coupling experiments. We also specify that the GenCP framework is architecture-independent and the model predictions are stable.
>
> > W1: One of the central motivation of the paper is that the training data generation using decoupled systems only is cheaper but this was unfortunately never quantified. The reviewer recommends evaluating the cost of training data generation for coupled and decoupled generation.
>
> A：We thank the Reviewer for pointing out this important missing quantification. Here we provide the following detailed instruction based on numerical solver principles, which has been added to the revised manuscript.
>
> Mainstream solvers for generating FSI data are generally categorized into monolithic solvers and partitioned solvers. To provide a theoretical and universal comparison (independent of specific implementations), we analyze the computational complexity of these two approaches for generating coupled data versus decoupled data.
>
> Monolithic solvers solve the governing equations of the fluid and solid simultaneously. Enforcing exact coupling conditions (equality of forces, displacements, and velocities) requires computing complex coupling terms, such as cross-derivatives in Jacobian matrices within global Newton iterations [1]. This enforces a computational burden much higher than solving two independent single-physics systems.
>
> Partitioned solvers are more commonly employed for FSI simulation. However, due to the temporal lag between fluid and solid solutions, maintaining stability is challenging. To prevent divergence, solvers must perform $K$ sub-iterations (coupling steps) within every single physical time step to achieve convergence. According to established studies [2, 3], the value of $K$ typically ranges from 5 to 20, and can exceed 100 in scenarios with strong added-mass effects. Consequently, generating one step of coupled data costs approximately $K \cdot (Cost_{fluid} + Cost_{solid})$. In contrast, our decoupled approach requires only 1 pass for each field per step. This implies that **coupled data generation is theoretically at least 5 times more expensive than decoupled generation**, even when ignoring the overhead of data transfer between solvers.
>
> Beyond raw computational time, we also emphasize the costs associated with human effort and stability. Coupled simulations are highly sensitive to hyperparameters and prone to divergence. Conversely, decoupled simulations, which treat interactions as fixed boundary conditions, are inherently robust and stable. This stability significantly reduces the effort required for data generation by eliminating the need for repeated trial-and-error debugging.
>
> Reference:
>
> [1] Hron, Jaroslav, and Stefan Turek. "A monolithic FEM/multigrid solver for an ALE formulation of fluid-structure interaction with applications in biomechanics." *Fluid-Structure Interaction: Modelling, Simulation, Optimisation*. Berlin, Heidelberg: Springer Berlin Heidelberg, 2006. 146-170.
>
> [2] Degroote, Joris, et al. "Stability of a coupling technique for partitioned solvers in FSI applications." *Computers & Structures* 86.23-24 (2008): 2224-2234.
>
> [3] Causin, Paola, Jean-Frédéric Gerbeau, and Fabio Nobile. "Added-mass effect in the design of partitioned algorithms for fluid–structure problems." *Computer methods in applied mechanics and engineering* 194.42-44 (2005): 4506-4527.

---

> ### Author Response · Authors · 2025-11-22
> **Official Response to Reviewer Uo4f (2)**
>
> > W2: The comparisons to baselines should also include frameworks trained on the fully coupled data to estimate the change in accuracy by using decoupled data only.
>
> A: Thank you for your reminder.  Actually, **the "Joint Training" row of** **Table 3 in the original submission has already presented the results of coupled training**, and we have also supplemented the results of Joint Training in Table 2 in the revised manuscript. As shown in the table, the error resulting from directly training on coupled data is obviously lower than that of methods using decoupled training for coupled inference. This is fully expected, as the latter approach represents a compromise when coupled data is unavailable. Actually, the model jointly trained on coupled data should represent the theoretical performance upper bound, which all decoupled training-based paradigms strive to approximate.
>
> | Field | $u$ | $v$ | $p$ | SDF |
> | --- | --- | --- | --- | --- |
> | Test Error on Turek-Hron Setting with Joint Training | 0.0088 | 0.0344 | 0.0544 | 0.0079 |

---

> ### Author Response · Authors · 2025-11-22
> **Official Response to Reviewer Uo4f (3)**
>
> > W3: The authors only apply their method to, in the setting of FSI, small problems. However, as stated in the Appendix for one of their examples the authors do not use decoupled data only but coupled training data as the decoupled data contained no meaningful structural motion. According to the Reviewer, this undermines one of the central statement of the paper. If already for a small example, using decoupled training data only is insufficient, the Reviewer would expect for more complex system with more non-linear coupling the same issue. This would significantly limit the applicability of the new approach.
>
> A: We thank the Reviewer for this valuable comment regarding the use of "half-decoupled" data in our experiments and the associated concern about our method’s capability to more complex coupled systems. We address this concern through both **theoretical analysis and new supplementary experiments**.
>
> First, we wish to clarify that the "half-decoupled" experimental design does not undermine our core algorithmic contribution about conditional-to-joint probabilistic sampling. As formalized by the error bound in Theorem 3.1, the total error of our generated joint distribution consists of two distinct components: the splitting error $(\tau)$ arising from the probability density evolution, and the learning error $(\epsilon)$ representing the approximation gap between learned and true conditional vector fields. The use of half-decoupled data in the original experiments merely simplified the learning task for $\epsilon_g$ (the structural field), but it did not alter or simplify the splitting mechanism $\tau$, which represents the core innovation of our algorithm. Therefore, **the theoretical validity of our "conditional-to-joint" sampling paradigm remains intact** regardless of the specific data complexity used for demonstration.
>
> Next, to empirically demonstrate that GenCP excels on strictly decoupled data even when applied to much **more complex systems with strong nonlinear coupling**, we have conducted **additional experiments** on a Nuclear-Thermal (NT) Coupling dataset as is used in [1]. As detailed in the newly added Appendix F, this problem involves three distinct physical fields (neutron, fluid, and solid), featuring highly complex interactions including regional and interface coupling, strong and weak coupling, bidirectional and unidirectional coupling. Specifically, this scenario is highly complicated, involving  solving neutron physics, heat conduction, and flow heat transfer equations, taking into account negative feedback between neutron physics and temperature, unidirectional coupling from fluid to neutron field, and strong interface coupling between solid and fluid. Our objective was to train models using strictly fully decoupled data from these three isolated fields and then infer the coupled physical trajectory of the entire system.
>
> For this task, we **conducted evaluations using GenCP, M2PDE, and surrogate-based paradigms**. The quantitative results, including validation errors on decoupled data and testing errors on coupled data, are summarized in the table below. **Detailed experimental results and additional visualizations are provided in Appendix F** in the revised manuscript.
>
> | Rel L2 Norm | Validation on decoupled data | Validation on decoupled data | Validation on decoupled data | Test on coupled data | Test on coupled data | Test on coupled data |
> | --- | --- | --- | --- | --- | --- | --- |
> | Field | Neutron | Solid | Fluid | Neurton | Solid | Fluid |
> | GenCP-FNO* | 0.0022 | 0.0006 | 0.0038 | **0.0085** | **0.0364** | **0.0270** |
> | Surrogate-FNO* | 0.0086 | 0.0014 | 0.0032 | 0.0149 | 0.0576 | 0.1095 |
> | M2PDE-FNO* | 0.0052 | 0.0014 | 0.0018 | 0.0136 | 0.1237 | 0.0463 |
> | GenCP-CNO | 0.0024 | 0.0005 | 0.0110 | **0.0044** | **0.0105** | **0.0330** |
> | Surrogate-CNO | 0.0046 | 0.0007 | 0.0082 | 0.0130 | 0.0553 | 0.3086 |
> | M2PDE-CNO | 0.0053 | 0.0016 | 0.0092 | 0.0164 | 0.0646 | 0.0401 |
>
> As shown in the table, GenCP **consistently outperforms** both the M2PDE and surrogate-based baselines across the neutron, fluid, and solid fields. Specifically, using FNO* as the backbone, GenCP achieves an average error reduction of 49.9% relative to M2PDE and 51.7% relative to the surrogate-based paradigm. With the CNO backbone, the improvements are even more substantial, with error reductions of 58.2% over M2PDE and 78.8% over surrogate-based paradigm. These results confirm that GenCP is **not limited by the simplified data settings used in the initial demonstration but is capable to predict complex, coupled dynamics from purely decoupled priors**.
>
> We believe the combination of theoretical clarification and new experiment can fully address reviewer's the concern.
>
> Reference:
>
> [1] Zhang, T., Liu, Z., Qi, F., Jiao, Y., and Wu, T. "M2PDE: Compositional generative multiphysics and multi-component PDE simulation." Proceeings of the Forty-Second International Conference on Machine Learning (2025).

---

> ### Author Response · Authors · 2025-11-22
> **Official Response to Reviewer Uo4f (4)**
>
> > W4: The obtained relative errors on the coupled data test set seem relatively high and would be too high for most engineering applications. The Reviewer acknowledges that the authors perform better than reported baselines but the gap between results on the decoupled validation set and the coupled test set seems to high and would discourage currently training on decoupled data in the Reviewers opinion. Moreover, the Reviewer encourages the authors to add the aforementioned baselines on a fully coupled dataset.
>
> A: Thank you for your valuable comment. Actually, we respectfully argue that the error magnitudes remain within a reasonable range and are not prohibitive. Actually, for a great deal of SOTA work in this field [1-3], the reported relative L2 norm errors are basically on the order of 1e-2. Moreover, we have successfully achieved **long-term prediction and coupled inference using decoupled training**. Ensuring reasonable accuracy in this setting is a significant challenge, while our method **successfully made it true**.
>
> Furthermore, even if the errors are considered slightly high, the paradigm of 'decoupled training for coupled inference' **still holds significant value in scenarios where coupled data is missing or unavalible**, which is widely acknowledged by the community (as detailed in response to W1).
>
> Regarding the Reviewer's suggestion to add baselines training on coupled data, we wish to clarify that our proposed paradigm is primarily about **inference** rather than **training**. In fact, in the context of training on coupled data (which is just an end-to-end prediction problem), the comparison between the baselines and our Gencp **reduces to the standard distinction** between surrogate models, diffusion models, and flow matching. Therefore, we believe such an experiment offers limited value for our contributions.
>
> Reference:
>
> [1] Kassaï Koupaï, A., Le Boudec, L., Serrano, L., and Gallinari, P. "ENMA: Tokenwise Autoregression for Continuous Neural PDE Operators." Proceedings of the Thirty-ninth Annual Conference on Neural Information Processing Systems (2025).
>
> [2] Serrano, L., Kassaï Koupaï, A., Wang, T. X., ERBACHER, P., and Gallinari, P. "Zebra: In-context generative pretraining for solving parametric PDEs." Proceedings of the Forty-second International Conference on Machine Learning (2025).
>
> [3] Wang, H., Xin, H., Wang, J., Yang, X., Zha, F., Dong, H., and Jiang, Y. "Mixture-of-experts operator transformer for large-scale PDE pre-training." Proceedings of the Thirty-ninth Annual Conference on Neural Information Processing Systems (2025).
>
> > W5: The authors use convolutional building blocks and do therefore require a regular grid. For FSI which often involves complex boundaries this is a significant limitation. In Figure 6, the cylinder is not round but the discretization artifacts are clearly visible. The authors should comment on this limitation.
>
> A：We appreciate the Reviewer's comment on geometric discretization. While our current implementation of the backbone architecture (CNO/FNO) is on regular grids, it is important to note that this is a limitation of the chosen backbone architectures, not the GenCP framework. GenCP is fundamentally **architecture-agnostic and representation-agnostic**. We employed regular grids to adhere to standard benchmarking practices in the "ML for Physics" domain, ensuring that our evaluation focused on the our proposed coupling paradigm itself. GenCP is fully compatible with mesh-based or point-cloud-based architectures (such as GNNs), making it capable of handling complex boundaries in engineering applications.
>
> Additionally, the visual artifacts noted are solely due to the resolution selected for computational efficiency. These artifacts would naturally resolve with finer discretizations, requiring no changes to our proposed coupling algorithm. We have added these to the discussion section of the revised manuscript as suggested.
>
> In summary, we want to emphasize that these limitations **do not undermine** the primary innovations and contributions of our proposed GenCP.

---

> ### Author Response · Authors · 2025-11-22
> **Official Response to Reviewer Uo4f (5)**
>
> > Q1: For the two FSI examples, the authors just report single error values but the Reviewer would have expected a distribution due to the probabilistic approach. Are these posterior mean values ?
>
> A：We thank the Reviewer for this accurate observation. Indeed, as a probabilistic modeling framework, our method inherently involves stochasticity during the sampling process. The errors currently reported in the manuscript are from one single sampling trajectory, which is consistent with prior studies [1-3].
>
> To further address your question, we have **additionaly computed error statistics** across 100 distinct samples generated from independent random noise, as summarized in the table below. The results demonstrate that, despite the generative nature of the model, GenCP successfully learns to produce **highly stable and deterministic outputs for this deterministic physical system**. Consequently, we consider the single-sample evaluation adopted in our paper to be reasonable and representative.
>
> | Field\Metric | Mean | Std Dev | Variance | 5th percentile | 95th percentile | Min | Max | Median |
> | --- | --- | --- | --- | --- | --- | --- | --- | --- |
> | Neutron Field | 0.008489 | 0.000065 | 4.177125e-09 | 0.008411 | 0.008578 | 0.008334 | 0.008828 | 0.008482 |
> | Solid Field | 0.036433 | 0.000036 | 1.326743e-09 | 0.036373 | 0.036483 | 0.036340 | 0.036510 | 0.036439 |
> | Fluid Field | 0.027038 | 0.000108 | 1.155901e-08 | 0.026880 | 0.027215 | 0.026790 | 0.027272 | 0.027033 |
>
> Reference:
>
> [1] Huang, J., Yang, G., Wang, Z., and Park, J. J. "DiffusionPDE: Generative PDE-solving under partial observation." Advances in Neural Information Processing Systems 37 (2024): 130291-130323.
>
> [2] Li, X., Zhang, J., Zhu, Q., Zhao, C., Zhang, X., Duan, X., and Lin, W. "From Fourier to Neural ODEs: Flow matching for modeling complex systems." Proceedings of the 41st International Conference on Machine Learning 235 (2024): 29390-29405.
>
> [3] Zhang, T., Liu, Z., Qi, F., Jiao, Y., and Wu, T. "M2PDE: Compositional generative multiphysics and multi-component PDE simulation." Forty-second International Conference on Machine Learning (2025).

---

### Official Review · Reviewer_sgcu · 2025-11-01

**Soundness:** 3
**Presentation:** 3
**Contribution:** 3
**Rating:** 6
**Confidence:** 4

**Summary:**

This paper proposes a novel and elegant generative paradigm—GenCP—for coupled multiphysics simulations. By formulating the coupled physics modeling problem as a probabilistic modeling problem, the authors' key innovation lies in combining probability density evolution from generative modeling with iterative multiphysics coupling. This enables training on decoupled data and inference of coupled physics processes during sampling. The authors evaluate their paradigm on a synthetic dataset and two challenging fluid-structure interaction scenarios to demonstrate GenCP's fundamental insights and superior application performance. This is an article about applying AI to engineering physics.

**Strengths:**

Combining probability density evolution with iterative multiphysics coupling in generative modeling is a novel approach, and the problem setting is very clear.

**Weaknesses:**

The theoretical part of this paper is excellent, but I think the experimental part could be scaled up further.

**Questions:**

Have you considered a larger-scale experiment?

---

> ### Author Response · Authors · 2025-11-22
> **Official Response to Reviewer sgcu (1)**
>
> First of all, we would like to thank you for recognizing the innovativeness of our method. Secondly, regarding your question about whether we can conduct **larger-scale experiments**, we have **added an additional experiment in a more complex coupling scenario**. Our method has achieved excellent results in this larger-scale scenario.
>
> > W1: The theoretical part of this paper is excellent, but I think the experimental part could be scaled up further.
> >
> > Q1: Have you considered a larger-scale experiment?
> >
>
> A: Thanks to the reviewer's suggestion on a larger-scale experiment. To empirically demonstrate that GenCP **scales effectively** to systems involving additional physical fields, stronger nonlinear couplings, and greater physical complexity, we have conducted **additional experiments** on a Nuclear-Thermal Coupling dataset.
>
> As detailed in the newly added Appendix F, this problem involves three distinct physical fields (neutron, fluid, and solid), featuring highly complex interactions including regional and interface coupling, strong and weak coupling, bidirectional and unidirectional coupling. Specifically, this scenario is **highly complicated**, involving  solving neutron physics, heat conduction, and flow heat transfer equations, taking into account negative feedback between neutron physics and temperature, unidirectional coupling from fluid to neutron field, and strong interface coupling between solid and fluid. Still, our objective was to train models using decoupled data and then infer transient evolution of this compelx system. We conducted evaluations using GenCP, M2PDE, and surrogate-based paradigms. The quantitative results, including validation errors on decoupled data and testing errors on coupled data, are summarized in the table below. Detailed experimental results and additional visualizations are provided in Appendix F in the revised manuscript.
>
> | Rel L2 Norm | Validation on decoupled data | Validation on decoupled data | Validation on decoupled data | Test on coupled data | Test on coupled data | Test on coupled data |
> | --- | --- | --- | --- | --- | --- | --- |
> | Field | Neutron | Solid | Fluid | Neurton | Solid | Fluid |
> | GenCP-FNO* | 0.0022 | 0.0006 | 0.0038 | **0.0085** | **0.0364** | **0.0270** |
> | Surrogate-FNO* | 0.0086 | 0.0014 | 0.0032 | 0.0149 | 0.0576 | 0.1095 |
> | M2PDE-FNO* | 0.0052 | 0.0014 | 0.0018 | 0.0136 | 0.1237 | 0.0463 |
> | GenCP-CNO  | 0.0024 | 0.0005 | 0.0110 | **0.0044** | **0.0105** | **0.0330** |
> | Surrogate-CNO | 0.0046 | 0.0007 | 0.0082 | 0.0130 | 0.0553 | 0.3086 |
> | M2PDE-CNO | 0.0053 | 0.0016 | 0.0092 | 0.0164 | 0.0646 | 0.0401 |
>
> As shown in the table, GenCP consistently outperforms both the M2PDE and surrogate-based baselines across the neutron, fluid, and solid fields. Specifically, using FNO* as the backbone, GenCP achieves an average error reduction of 49.9% relative to M2PDE and 51.7% relative to the surrogate-based paradigm. With the CNO backbone, the improvements are even more substantial, with error reductions of 58.2% over M2PDE and 78.8% over surrogate-based paradigm. These results demonstrate that our method **maintains superior** performance even when **scaled to more complex** physical systems.

---

### Official Review · Reviewer_k4vA · 2025-11-05

**Soundness:** 3
**Presentation:** 4
**Contribution:** 3
**Rating:** 6
**Confidence:** 4

**Summary:**

The submission proposed a flow-based model to handle multiphysics simulation. The idea is simply: training two flow models independently but using another variables as the condition. It reduces the training complexity. I appreciate the method, but not satisfied with current experiments part.

**Strengths:**

I think the proposed method is easy to follow and the result is good. Also, I think the problem the authors are trying to solve is meaningful.

**Weaknesses:**

1. I think the experiments are not enough. It all concentrates on fluid dynamics, but it is better to combine fluid and rigid objects. It is convenient to treat the whole fluid system as a domain.

2. Also for the experiments part, I feel like the submission should add a baseline: training a single v(x_t, y_t) for all variables (coupled data), to prove the decouple the data is beneficial.

**Questions:**

Line 202, Line 207, I believe in the flow model, the vf = f1-f0, not f1-zf, the same for vg.

---

> ### Author Response · Authors · 2025-11-22
> **Official Response to Reviewer k4vA (1)**
>
> We appreciate your recognition that the problem we aim to solve is meaningful, as well as your pointing out the shortcomings of our experiments. Therefore, we have made important supplements to the manuscript. First, we have **added a more complex and realistic Nuclear-Thermal Coupling scenario** and **supplemented a joint-training experiment**. Second, we have provided clarifications about formula symbols.
>
> > W1: I think the experiments are not enough. It all concentrates on fluid dynamics, but it is better to combine fluid and rigid objects. It is convenient to treat the whole fluid system as a domain.
>
> A：Thanks for pointing out the shortcomings of our experiment. It is true that the fluid-structure interaction problem is relatively easy to handle, and our fluid system does not take into account an overly complex solid field either. Therefore, we have **supplemented a new experiment** on Nuclear-Thermal Coupling setting as described in Appendix F in the revised manuscript, which involves three coupled fields, including neutron, fuel and fluid. This is a **highly complicated scenario**, involving solving neutron physics, heat conduction, and flow heat transfer equations, taking into account negative feedback between neutron physics and temperature, unidirectional coupling from fluid to neutron field, and strong interface coupling between solid and fluid.
>
> We conducted evaluations using GenCP, M2PDE, and surrogate-based paradigms. The quantitative results, including validation errors on decoupled data and testing errors on coupled data, are summarized in the table below.
>
> | Rel L2 Norm | Validation on decoupled data | Validation on decoupled data | Validation on decoupled data | Test on coupled data | Test on coupled data | Test on coupled data |
> | --- | --- | --- | --- | --- | --- | --- |
> | Field | Neutron | Solid | Fluid | Neurton | Solid | Fluid |
> | GenCP-FNO* | 0.0022 | 0.0006 | 0.0038 | **0.0085** | **0.0364** | **0.0270** |
> | Surrogate-FNO* | 0.0086 | 0.0014 | 0.0032 | 0.0149 | 0.0576 | 0.1095 |
> | M2PDE-FNO* | 0.0052 | 0.0014 | 0.0018 | 0.0136 | 0.1237 | 0.0463 |
> | GenCP-CNO | 0.0024 | 0.0005 | 0.0110 | **0.0044** | **0.0105** | **0.0330** |
> | Surrogate-CNO | 0.0046 | 0.0007 | 0.0082 | 0.0130 | 0.0553 | 0.3086 |
> | M2PDE-CNO | 0.0053 | 0.0016 | 0.0092 | 0.0164 | 0.0646 | 0.0401 |
>
> As shown in the table, GenCP consistently outperforms the baselines across three fields. Specifically, using FNO* as the backbone, GenCP achieves an average error reduction of 49.9% relative to M2PDE and 51.7% relative to the surrogate-based paradigm. With the CNO backbone, the improvements are even more substantial, with error reductions of 58.2% over M2PDE and 78.8% over surrogate-based paradigm. These results confirm that GenCP is **not limited** by your mentioned "easy fluid dynamics" but is **capable to model complex, coupled physical systems**. We hope this supplemented experiment with more complex settings can address your concern.
>
> > W2: Also for the experiments part, I feel like the submission should add a baseline: training a single v(x_t, y_t) for all variables (coupled data), to prove the decouple the data is beneficial.
>
> A：Thank you for your reminder. The row of "Joint Training" of Table 3 in **our original submission has already presented** the results of coupled training, which is exactly what you referred to as "training a single $v(x_t, y_t)$ for all variables (coupled data)". Meanwhile, we have also **supplemented the joint training results** in Table 2 in the revised manuscript, just as follows. It can be seen that compared with the baseline, our method achieves a smaller reduction in error than the results trained on coupled data.
>
> | Field | $u$ | $v$ | $p$ | SDF |
> | --- | --- | --- | --- | --- |
> | Test Error on Turek-Hron Setting with Joint Training | 0.0088 | 0.0344 | 0.0544 | 0.0079 |
>
> > Q1: Line 202, Line 207, I believe in the flow model, the vf = f1-f0, not f1-zf, the same for vg.
>
> A：Thank you for pointing out the confusion. In fact, in our paper, $f_0$ refers to the initial condition of physical evolution and also serves as the condition for the flow model. Meanwhile, $z_f$ is a sample from the initial distribution of the flow model. Therefore, the flow model learns the velocity field from $z_f$ to $f_1$, and the formulas in the paper are correct. We hope this addresses your concern.

---

### Author Response · Authors · 2025-11-22
**General Response**

We thank all reviewers for their thorough evaluations and constructive suggestions. We appreciate the positive comments acknowledging the **significance of our problem setting** (Reviewers k4vA, sgcu, Uo4f, CPwt), the **strength and rigor of our theoretical formulation** (Reviewers sgcu, Uo4f, CPwt), the **clarity and completeness of our exposition** (Reviewers k4vA, CPwt), and the our **contribution to practical scientific computing** (Reviewers Uo4f, CPwt). We have revised the manuscript substantially and conducted new experiments to address all concerns raised by the reviewers. The modifications have been highlighted in blue in the revised version.

Motivated by the reviewers’ valuable feedback, we have made the following key additions and improvements:

1. **A new large-scale 3-field Nuclear–Thermal Coupling experiment has been added (Appendix F in the revised manuscript).** In this complicated and strongly coupled multiphysics system, GenCP achieves 50%–80% lower error than baselines. This addresses concerns about fully-decoupled data (Reviewers Uo4f, CPwt), simplicity of previous benchmarks (Reviewer k4vA) and scalability to more complex systems (Reviewers sgcu, Uo4f, CPwt).
2. **Inclusion of joint-training baselines** in both Table 2 and Table 3, which represents the upper bounds of infering on coupled data with decoupled training. This experiment responds directly to requests from Reviewers k4vA and Uo4f.
3. **Comparison of data-generation cost** between decoupled and coupled simulations, clarifying why decoupled data is substantially cheaper and easier to obtain (Reviewer Uo4f and CPwt).
4. **A clearer explanation of theoretical intuition**, including detailed clarifications on continuity equations in function space, Lie–Trotter splitting, and the role of conditional dynamics in reconstructing coupled behavior (Reviewers CPwt, k4vA).
5. **The insight of the key for reconstructing coupled dynamics from decoupled data** addresses Reviewer CPwt’s conceptual questions and proposes instruction for general apllication of our paradigm.
6. **Probabilistic-model stability evaluation.** We perform 100 independent samplings and report full statistics, demonstrating that the generative dynamics remain highly deterministic, which addresses Reviewer Uo4f's concern regarding the reported error from single sampling.

We believe these new experiments, expanded analyses, and improved theoretical explanations directly address all reviewers' concerns and further strengthen our contributions. We sincerely thank the reviewers for their insightful comments and hope the revisions meet their expectations.

---

### Author Response · Authors · 2025-12-02
**Summary of the Discussion Period**

Dear Area Chair,

We sincerely appreciate the considerable time and effort you have devoted to re-evaluating the rebuttal and discussion of our manuscript. We provide this concise summary to assist in your final assessment, demonstrating that we have comprehensively addressed the reviewers' concerns.

**Novelty and Contribution:**

As acknowledged by the reviewers, GenCP introduces a **novel, principled, and elegant "conditional-to-joint" generative paradigm**. It allows for **learning coupled physics strictly from decoupled data**, thereby addressing a major data scarcity bottleneck in multiphysics simulation. Furthermore, by theoretically bridging **Probability Density Evolution (Flow Matching)** with **Operator Splitting**, we provide rigorous proof of the error controllability of our method.

**Rebuttal Status:**

Currently, Reviewer CPwt has explicitly **confirmed that their concerns have been "fully resolved"**. While other reviewers have not yet responded to our latest updates, we have **completely addressed each of their concerns, suggestions, and questions, just as outlined in our General Response**.
We thank the reviewers for their constructive input, which has ultimately strengthened our paper. We are confident that the revised manuscript, supported by the new experiments and clarifications, stands as a solid and impactful contribution to the ICLR community.

Best regards,

Authors of ICLR Submission 2289

---

### Meta-Review · Area_Chair_FuYj · 2025-12-24

**Summary:**

Reviewers k4vA and sgcu left short, uninformative reviews which are not worth considering.

The other reviewers raised a number of potential concerns: the experiments did not compare against fully coupled data; it was unclear if uncoupled data generation is truly cheaper; decoupled data is sometimes uninformative and led to the introduction of a "half-decoupled" approach which could undermine the message of the paper; the experiments could be strengthened; the magnitude of the errors were large, and error bars were not reported.

In each case, I believe the authors have addressed the concerns clearly, by providing further justification and references, rewriting of the paper, and further experimental validation. The overall scores are borderline, but likely would have crossed the threshold to acceptance during the rebuttal phase. Therefore, this paper merits acceptance.

**Reviewer Concerns:**

See above; I believe all concerns were addressed to an acceptable extent.

**Reviewer Scores:**

Reviewers k4vA, sgcu: The original reviewers were uninformative and I think it is unlikely that these reviewers would have engaged in the discussion.

Reviewer Uo4f: Since the authors carefully addressed each of the reviewer's concerns, I think the reviewer would have raised the score at least to borderline.

Reviewer CPwt: This reviewer's initial score was already high, and the reviewer is satisfied by the rebuttal.

---

### Decision · Program_Chairs · 2026-01-26

Accept (Poster)